# Relaxed selection underlies genome erosion in socially parasitic ant species

Lukas Schrader [1,2✉], Hailin Pan[3], Martin Bollazzi [4], Morten Schiøtt [1], Fredrick J. Larabee [5], Xupeng Bi[3], Yuan Deng[3], Guojie Zhang [1,3,6], Jacobus J. Boomsma [1,9✉] & Christian Rabeling [7,8,9✉]

Inquiline ants are highly specialized and obligate social parasites that infiltrate and exploit colonies of closely related species. They have evolved many times convergently, are often evolutionarily young lineages, and are almost invariably rare. Focusing on the leaf-cutting ant genus *Acromyrmex*, we compared genomes of three inquiline social parasites with their free-living, closely-related hosts. The social parasite genomes show distinct signatures of erosion compared to the host lineages, as a consequence of relaxed selective constraints on traits associated with cooperative ant colony life and of inquilines having very small effective population sizes. We find parallel gene losses, particularly in olfactory receptors, consistent with inquiline species having highly reduced social behavioral repertoires. Many of the genomic changes that we uncover resemble those observed in the genomes of obligate non-social parasites and intracellular endosymbionts that branched off into highly specialized, host-dependent niches.

[1] Centre for Social Evolution, Department of Biology, University of Copenhagen, Copenhagen, Denmark. [2] Institute for Evolution and Biodiversity, University of Münster, Münster, Germany. [3] BGI-Shenzhen, Shenzhen, China. [4] Entomología, Facultad de Agronomía, Universidad de la República, Montevideo, Uruguay. [5] Department of Entomology, National Museum of Natural History, Smithsonian Institution, Washington, DC, USA. [6] State Key Laboratory of Genetic Resources and Evolution, Kunming Institute of Zoology, Chinese Academy of Sciences, Kunming, China. [7] Department of Biology, University of Rochester, Rochester, NY, USA. [8] School of Life Sciences, Arizona State University, Tempe, AZ, USA. [9] These authors contributed equally: Jacobus J. Boomsma, Christian Rabeling. ✉email: Lukas.Schrader@wwu.de; jjboomsma@bio.ku.dk; christian.rabeling@asu.edu

Radical shifts to specialized ecological niches change selection regimes often resulting in fundamental genomic and phenotypic transformation[1,2]. Selection on traits adapted to the ancestral niche will relax and positive selection will favor adaptations to new ecological settings and matching life histories[3–5]. In addition, niche specialization can lead to small isolated populations with limited gene flow[6–8], decreasing the efficiency of selection[9]. Recent research revealed that consequences of relaxed selection are partly predictable, both at a general level and at the level of particular traits[10].

The consequences of relaxed selection can be dramatic, e.g., in the convergent regression of eyesight in animals that shifted from a diurnal habitat to permanent darkness. Cave fish[11], and subterranean mammals[12] rapidly lost or reduced vision, consistent with eyes as complex organs no longer being maintained by stabilizing selection[13]. Chloroplasts and the genetic capacity for photosynthesis degenerated in plants that transitioned to a holoparasitic lifestyle[14,15], and genomes and body plans of parasitic Myxozoa are significantly reduced compared to those of free-living cnidarians[16]. Similar changes are also known from other obligate parasites[17–20]. Finally, evolutionary transitions in bacteria from a free-living to an endosymbiotic lifestyle are almost invariably associated with convergent phenotypic and genomic reductions[21]. In many of these cases, changes in effective population sizes (Ne) due to permanent asexual reproduction or frequent bottlenecks reduce the efficacy of selection. This results in impoverished genomes with an excess of slightly deleterious alleles and only essential functions being actively maintained by selection[22]. Genomes of recently acquired endosymbionts have eroded significantly less than genomes of ancient endosymbionts[22] and similar trends apply to the nuclear and cytoplasmic genomes of parasitic plants and animals[23,24]. In general, genomic and phenotypic changes following niche specialization appear to often evolve surprisingly fast[25,26].

It is less generally known that there are also examples of social niche shifts that have induced rapid change in morphological and behavioral phenotypes, with the most extreme transitions being found in inquiline social parasites of free-living ants. These morphologically and behaviorally highly specialized species parasitize colonies of other ants by exploiting their hosts' social behavior such as brood care, food resources, and an established and protected nest environment[27,28]. Analogous to intracellular and specialized non-social parasites, the evolution of inquilines has classically been described as a degenerative process[29,30] almost invariably converging on a similar suite of morphological and behavioral characteristics across independently evolved inquilines[31]. This so-called "social parasite syndrome" subsumes an array of regressive phenotypic changes, such as the reduction or complete loss of the worker caste, simplified mouthparts, antennae and integuments, loss of certain exocrine glands, and a nervous system of reduced complexity likely associated with a drastically narrowed behavioral repertoire[27–30,32,33]. Populations of inquiline social parasites are almost invariably small and patchily distributed, dispersal distances are low, and mating often occurs between siblings close to or inside a host nest[27,28,34,35].

Inquiline social parasites have evolved convergently across six subfamilies of ants[28], and they are often their host species' closest relatives (Emery's rule)[36]. Unlike in non-social parasites and endosymbionts, a few well-studied cases have provided solid evidence for some inquilines having speciated directly from their hosts in sympatry[28,34,37,38]. The fungus-growing ant genus *Acromyrmex* is well-suited for comparative studies of social parasite evolution because it harbors five recently evolved species of inquilines (Table 1). Three of these, the Brazilian *Acromyrmex ameliae*[39], the Panamanian *Acromyrmex insinuator*[40], and the Uruguayan *Acromyrmex charruanus*[41], are morphologically less specialized than the other two, the South American *Pseudoatta argentina*[42] and the Brazilian *Acromyrmex fowleri*[43], both of which exhibit suits of behavioral and morphological traits that strictly adhere to the social parasite syndrome[27,43]. To explore whether the dynamics of evolutionary change in inquiline social parasites follow similar patterns as documented in non-social parasites, intracellular endosymbionts, and other lineages that underwent radical niche shifts, we obtained and analyzed full genome sequences of three *Acromyrmex* inquilines and their respective host species.

Our analyses confirm that inquiline lineages have accumulated convergently evolved signatures of genome-wide and trait-specific genetic erosion. In particular, we find that (1) natural selection in inquiline species is relaxed overall, most likely because many ancestral social traits have become redundant and because Ne, which is expected to be low in social insects per se[44], is further decreased in social parasites, (2) losses and reductions have particularly affected the olfactory receptor repertoire of social parasites, (3) genomic changes are most extreme in the highly modified social parasite with the most quintessential match to the inquiline syndrome, and (4) losses/reductions at the genome-level emerged through fixation of different kinds of genomic rearrangements. We infer that relaxed selection accelerated general genome erosion in social parasites and alleviated evolutionary constraints, which facilitated rapid adaptive evolution of specific traits associated with a socially parasitic lifestyle. Our findings advance our understanding of the genomic consequences of transitioning to a novel, highly specialized life history and provide detailed insights into the molecular evolution of social parasitism in ants.

## Results

To identify genetic changes associated with the shift from cooperative social colony life to social parasitism, we de novo sequenced genomes of four *Acromyrmex* species, including the three inquiline social parasites *A. charruanus*, *A. insinuator*, and *Pseudoatta argentina*, as well as *A. heyeri*, the host species of *A. charruanus* and *P. argentina*. The genome of *A. echinatior*, the host of *A. insinuator*, was published previously[45] and included in our comparisons. De novo assembly produced genomes of similar quality as previously published genomes of other ants[46]. Genome assembly completeness, assessed with BUSCO ("genome" mode), ranged between 91.4% for *A. charruanus* and 97.3% for *A. insinuator*, which is comparable to other published attine genome assemblies[47]. Protein-coding genes were annotated for the four newly sequenced genomes and reannotated for seven previously published attine genomes[47]. We annotated between 13,249 (*P. argentina*) and 16,193 (*A. heyeri*) protein-coding genes (median = 14,688) and retrieved 6338 single-copy orthologs. BUSCO scores for protein annotations ("protein" mode) were lower than for the genome assemblies, particularly for *P. argentina* (89.1% vs. 97.1%), but these differences could be accounted for in our analyzes of gene family size evolution (see below).

Time-calibrated phylogenetic analyses of orthologous fourfold degenerate sites inferred two independent origins of inquiline social parasitism in *Acromyrmex* (Fig. 1). First, a South American lineage where *A. heyeri* separated from the common (putatively socially parasitic) ancestor of *A. charruanus* and *P. argentina* before the two social parasites diverged and, second, a Central American speciation event when *A. insinuator* diverged from its host *A. echinatior*. Both origins of inquilinism are evolutionarily recent, estimated to be ~2.5 Ma for the divergence between *A. heyeri* and the common ancestor of *A. charruanus* and *P. argentina*, and ~1 Ma for the divergence between *A. insinuator* and *A. echinatior*. The two South American social parasites,

**Table 1 Morphological and life history traits characteristic of the socially parasitic ant inquiline syndrome in the five known inquiline species within the *Acromyrmex* leaf-cutting ants (modified from ref. [43]).**

| | *A. insinuator* | *A. ameliae* | *A. charruanus* | *A. fowleri* | *P. argentina* |
|---|---|---|---|---|---|
| Elongated antennal scapes relative to the host species | x | x | x | x | x |
| Multiple egg-laying social parasite queens coexist in a host colony | x | x | ? | x | |
| Social parasite queens peacefully coexists with the host queen | x | x | x | x | |
| Reduced queen body size relative to the host | x | x | x | x | x |
| Partial or complete loss of the inquiline worker caste | (x) | (x) | x | x | x |
| Shiny integument relative to opaque host queens | | | | x | x |
| Reduced pilosity | | | | x | x |
| Number of antennal segments reduced in the male | | | | | x |
| Number of maxillary palps reduced | | | | | x |
| Males resemble queens morphologically (gynaecomorphism) | | | | | x |
| Sib-mating inside the nest (adelphogamy) | | | | | x |

*A. insinuator, A. charruanus* and *P. argentina* were included in this study.

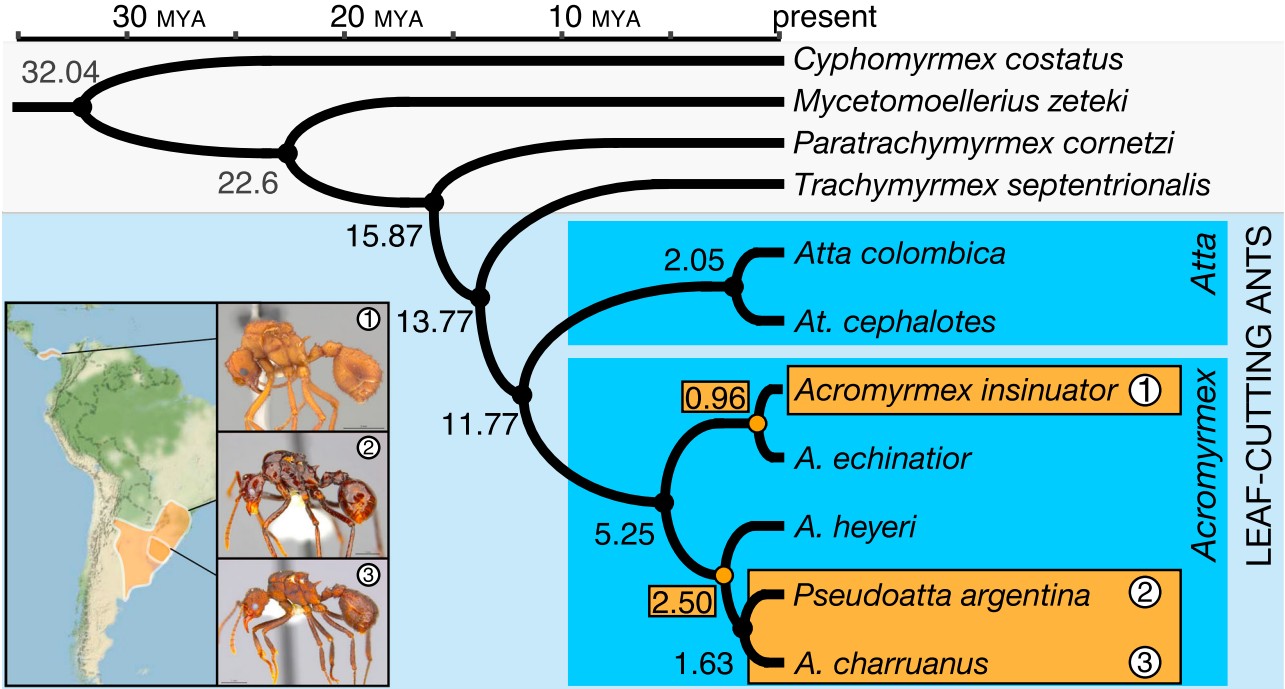

**Fig. 1 Divergence estimates for *Acromyrmex* host and inquiline parasite species.** Time-calibrated phylogeny of the fungus-growing ants for which genomes have been sequenced, including the three inquiline social parasite species and their two host species. The two origins of social parasitism in *Acromyrmex* (orange dots and boxes) occurred ca. 0.96 Ma ago for *A. insinuator* (1) and ca. 2.50 Ma ago when the ancestor of *A. heyeri* diverged from the stem group representative of *Pseudoatta argentina* (2) and *A. charruanus* (3).

*A. charruanus* and *P. argentina*, diverged ~1.6 Ma ago from their most recent common ancestor. Our analyses largely agreed with previously published divergence time estimates in the leaf-cutting ants, i.e., the divergence between *Acromyrmex* and *Atta* (~12 Ma, compared to 10–16 Ma[45,47,48]), but our present study suggests a more recent divergence of *At. cephalotes* and *At. colombica* (~2 Ma) than previously estimated (~7 Ma)[47].

**Genomic evidence of relaxed selection in social parasites.** All inquiline social parasite species are rare, patchily distributed, live in small populations, and likely to be inbred[27,35]. Inbreeding and small census population sizes are expected to significantly decrease the effective population sizes ($N_e$) of social parasites relative to their non-parasitic relatives[49,50]. Coalescent analyses on one host-parasite pair (*A. echinatior* and *A. insinuator*) and a third, non-parasitic, closely related species (*A. octospinosus*)[40] for

which the appropriate samples (two individuals from different colonies) could be collected, confirm that $N_e$ was indeed consistently smaller in the social parasite compared to its free-living relatives (Fig. 2A). Evolutionary rate analyses revealed genome-wide relaxation of selection on coding sequences across all socially parasitic lineages (Fig. 2B–E). Consistent with relaxed selection, codon usage was overall less biased in parasites (see Supplementary Information for details), although the signal was much weaker than at the level of non-synonymous substitution rates. Average $d_N/d_S$ ratios of single-copy orthologs were significantly increased in social parasite lineages relative to their non-parasitic sister lineages ($p_{Mann–Whitney} < 1e-5$, Fig. 2B), and selection intensity parameter $\log_2(k)$ analysis[51] showed this difference to be due to an overall preponderance of relaxed selection (Fig. 2C–E). We uncovered clear signatures of relaxed purifying selection (in coding regions with average $d_N/d_S < 1$, Fig. 2C) and

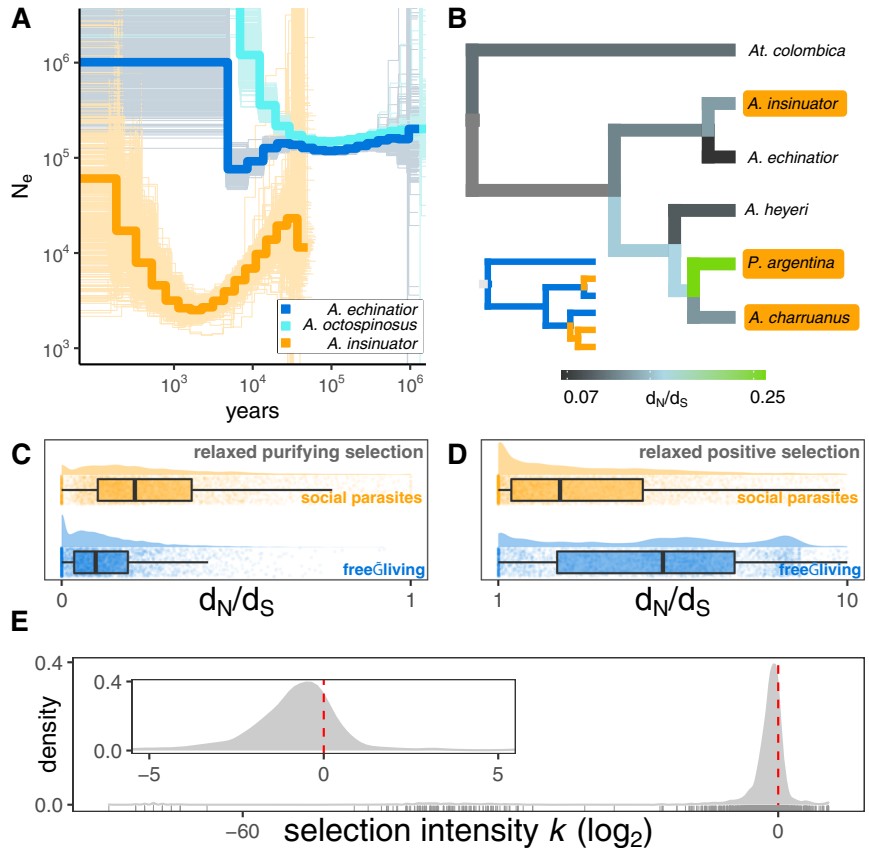

**Fig. 2 Effective population sizes and molecular evolution in inquiline social parasites and their related free-living *Acromyrmex* host species. A** Effective population size ($N_e$) of the social parasite *A. insinuator* is consistently smaller than of its close relatives *A. echinatior* and *A. octospinosus*. Solid blue and orange lines show average $N_e$ estimates across 500 bootstrap replicates (thin lines in the background). **B** Phylogram showing median rates of non-synonymous to synonymous substitutions ($d_N/d_S$) for 3850 single-copy orthologs (filtered for $d_N/d_S < 10$, see methods), with *At. colombica* as outgroup (representing the sister genus *Atta* in which no inquilines appear to have evolved) and a smaller inset tree showing only relationships between parasitic (orange) and free-living (blue) lineages. Average evolutionary rates were significantly increased in the socially parasitic lineages compared to their free-living host relatives. **C–D** Distribution of evolutionary rates for regions in 3616 single-copy orthologs across all species (filtered for $d_N/d_S < 10$, see Methods) under purifying selection ($d_N/d_S < 1$, **C**) and positive selection ($d_N/d_S \geq 1$, **D**). Boxplot centers show the median, hinges show the first and third quantiles, and whiskers show the 1.5× inter-quantile range. Evolutionary rates are significantly shifted towards $d_N/d_S = 1$ in social parasites (orange) relative to free-living lineages (blue) indicating relaxation of purifying selection in regions where $d_N/d_S < 1$ (**C**) and relaxation of positive selection ($d_N/d_S \geq 1$, **D**). Histograms and boxplots show inferred $d_N/d_S$ rate distributions for single-copy orthologs, and background dot plots show evolutionary rates for individual orthologs. **E** Distribution of the selection intensity parameter $\log_2(k)$[51] for single-copy orthologs in social parasite lineages. Positive values indicate an intensification of selection and negative values suggest a relaxation of natural selection in the inquiline social parasites relative to non-parasitic lineages. Selection intensity was highly negative ($\log_2(k) < -2$) in 792 genes, weakly negative ($-2 < \log_2(k) < 0$) for 2186 genes, weakly positive ($0 < \log_2(k) < 2$) for 563 genes and highly positive ($\log_2(k) > 2$) for 75 genes. The inset plot at the top left corner focuses on the narrower *x*-axis range of $-5 < \log_2(k) < 5$.

relaxed positive selection (where average $d_N/d_S > 1$, Fig. 2D). In both cases, average evolutionary rates in social parasites shifted towards $d_N/d_S$ ratios of 1, which is indicative of neutral evolution. Accordingly, the selection intensity parameter $\log_2(k)$ was weakly negative for most genes throughout these genomes, and strongly negative for a long-stretched tail end (Fig. 2E). However, there was also a minority of genes that experienced a modest intensification of selection as the peaks towards the right in Fig. 2E indicate. Our results are consistent with a prevailing pattern of genome-wide erosion across social parasite genomes punctuated by intensified selection on specific genes.

In addition to genome-wide signatures of relaxed selection, the transition from cooperative social colony life to exploitative social parasitism should particularly relax selection on genes coding for complex phenotypic traits associated with social behavior such as brood care, foraging, and nest construction. We identified 233 genes (3.34% of the tested genes) showing evidence of relaxed selection in at least one of the social parasite branches (at FDR < 0.1 and relaxation parameter $k < 1$) among single-copy orthologs (see Supplementary Figs. 14–16, Supplementary Table 34). However, we also detected signatures of intensified selection (at FDR < 0.1, $k > 1$) in 102 genes (1.46%, Supplementary Figs. 17–19, Supplementary Table 35), suggesting that the convergent niche shifts to social parasitism also triggered adaptive changes within an overall context of more genomic erosion.

**Gene loss in social parasites.** Gene content and synteny have generally evolved rapidly across the genomes of ants and of fungus-growing ants in particular[47,52,53]. In such generally dynamic background of change, we expected to see genomic signatures of trait loss of similar or larger magnitude than what is known for non-social animal and plant parasites[20,54]. To explore gene family evolution in social parasites, we modeled gene family size changes with CAFE, accounting for potential biases from

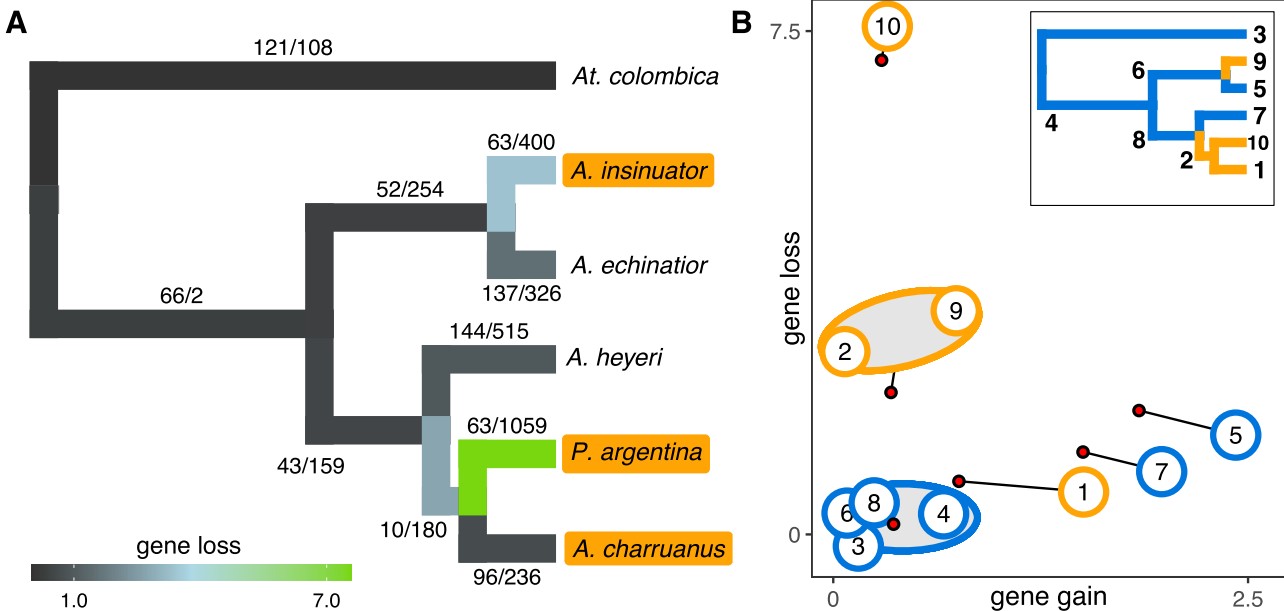

**Fig. 3 Evolution of gene family size in *Acromyrmex* leaf-cutting ants and their social parasites, with *At. colombica* as outgroup. A** The phylogenetic relationships of free-living and socially parasitic species with branches colored according to inferred frequencies of gene loss per gene per thousand years. Numbers along branches indicate inferred gene family expansions/contractions. **B** Rates of gene gain (x-axis) and gene loss (y-axis) for all different branches. Large numbered dots (1–10) show rates for individual branches inferred from 2-parameter models. Small red dots show results from a 6-parameter model, which grouped nodes 3, 4, 6, and 8 in one cluster that was unaffected by social parasitism and the two origins of social parasitism (the stem group of *A. charruanus* and *P. argentina* (node 2) and *A. insinuator* (node 9)) in another cluster characterized by much higher degrees of gene loss. The two hosts of the inquilines, *A. echinatior* (node 5) and *A. heyeri* (node 7), are both characterized by much higher rates of gene gain than the lineages that were never affected by social parasitism. Relative to their common ancestor (node 2), *P. argentina* presents as an outlier (node 10) characterized by massive gene loss without gene gain, while its sister inquiline *A. charruanus* reversed gene loss to a level reminiscent of leafcutter ants that were not affected by social parasitism. The deviant positions in 'gene-loss and gene-gain space' of all species involved in social parasitism suggest that antagonistic coevolution may have affected dynamics of genome-wide evolution across hosts and social parasites. The inset phylogram provides node numbers and highlights free-living lineages (blue) and parasitic lineages (orange).

incomplete assemblies and annotations. Models included *Acromyrmex* host and social parasite species, with *At. colombica* serving as outgroup, from which no inquilines are known. Our analysis showed that gene family evolution at three of the four social parasite nodes is indeed largely characterized by gene losses (Fig. 3A, B). We uncovered high rates of gene loss and low rates of gene gain in *P. argentina*, and also at the two origins of social parasitism (i.e., in *A. insinuator* and the ancestor of *A. charruanus* and *P. argentina*). Intriguingly, however, *A. charruanus* showed a different pattern, with the number of gene gains exceeding the number of gene losses, similar to the pattern observed in free-living attine species. This indicates that during the first ~1 Ma of inquiline evolution, *A. insinuator* and the common ancestor of *A. charruanus* and *P. argentina* followed a parallel pattern of predominantly gene loss, but that the later split between *A. charruanus* and *P. argentina* induced remarkably divergent trajectories in gene family size evolution.

We identified gene families characterized by high rates of gene loss in social parasite species compared to their hosts by functional enrichment analyses. This showed that gene loss had most significantly affected olfactory receptors (ORs), as only GO terms related to olfaction were significantly enriched (e.g., GO:0004984 "olfactory receptor activity", FDR = 5.6e−6, Supplementary Table 45). No GO terms were significantly enriched among genes gained in social parasites or genes lost in the host species. The odorant receptor gene family is directly relevant to the behavior of social insects because colony members are crucially dependent on efficient chemical communication via olfactory cues. It was recently documented that the OR gene

family experienced a significant expansion in ants[55,56]. Our comprehensive annotation and analysis of OR repertoires (Fig. 4, Table 2, see Supplementary Information for details) confirmed a parallel reduction in all three *Acromyrmex* social parasites, consistent with these species being less engaged in social interactions than their closely related host species. In *A. insinuator* we identified 32 fewer genes containing the characteristic 7tm6 domain of ORs than in its host *A. echinatior*. In the other lineage, we detected a comparable reduction of 38 fewer ORs in *A. charruanus* and a more massive reduction in *P. argentina* where only 263 OR genes remained compared to 467 OR genes in the host *A. heyeri*. This 44% loss implied that *P. argentina* approached the number of OR genes typical for nonsocial Hymenoptera such as *Nasonia vitripennis* with 225 ORs[57], consistent with *P. argentina* having the most highly specialized socially parasitic lifestyle of all known *Acromyrmex* inquilines[43,58] (Table 1). These results suggest that relaxed purifying selection on formerly important social traits has affected odorant receptor gene losses of social parasites in proportion to their degree of phenotypic specialization.

To gain a more detailed understanding of OR gene family evolution in fungus-growing ants, we reconstructed a family-wide gene tree, dividing genes into 321 clades belonging to 27 gene subfamilies (Fig. 4A). More than 30% of all OR clades (101 of 321) contained fewer paralogs in at least one parasite species compared to their respective hosts (orange highlights in Fig. 4A), with six clades being convergently reduced in the parasites relative to their hosts (orange number labels). In accordance, fewer large OR gene arrays (>8 genes) were retained in parasites

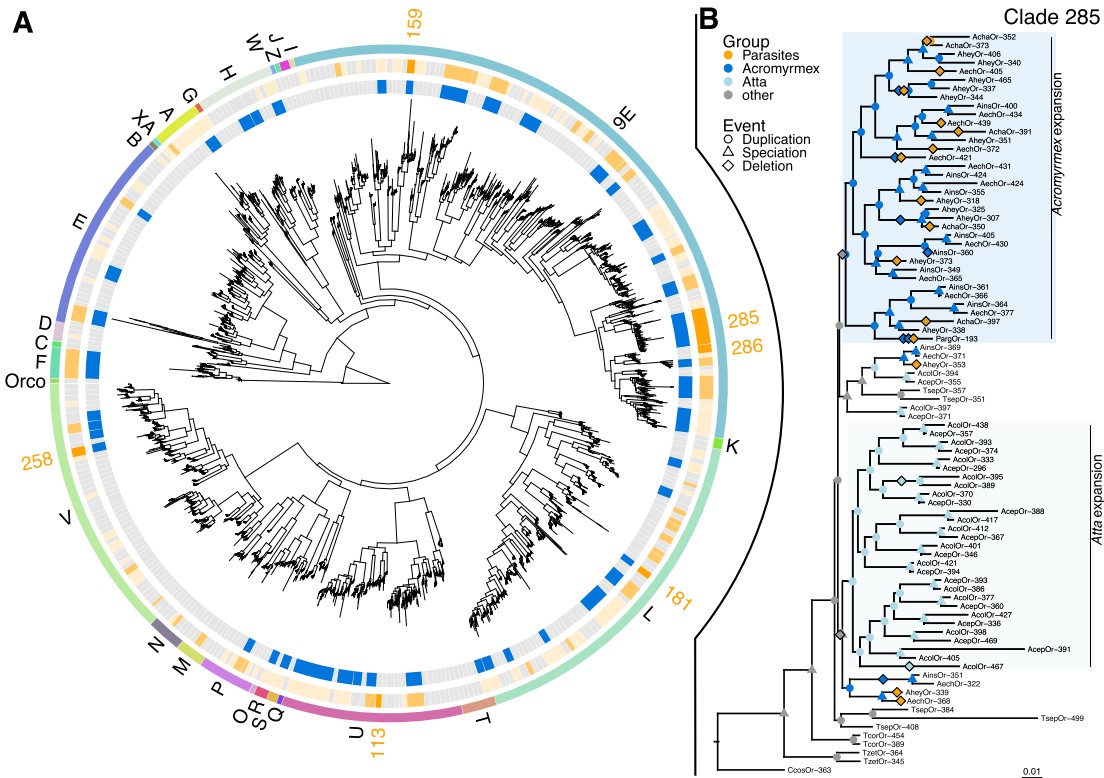

**Fig. 4 Odorant receptor (OR) gene family evolution in fungus-growing ants. A** Gene family tree for attine ant OR genes. The outermost circle shows gene subfamilies, the middle circle highlights OR gene clades with reductions in inquiline social parasites (orange blocks and numbers), and the inner circle highlights OR gene clades that expanded in the leaf-cutting ant species compared to the phylogenetically more basal non-leaf-cutting attine ants (blue blocks; cf. basal branches in Fig. 1). **B** Reconciled gene tree of OR clade 285 belonging to the 9-exon subfamily that diversified independently and at a particularly high rate in the two leaf-cutting ant genera and subsequently underwent convergent gene losses in all social parasites. Branches with inferred gene deletion events in inquiline social parasites and their hosts are marked in orange diamonds, duplications are marked with circles colored according to the taxonomic affiliation of the ants, and speciation nodes are marked with triangles. Background shading colors highlight the independent expansions in *Atta* and *Acromyrmex*.

**Table 2 Gene family sizes in *Atta* and *Acromyrmex* leaf-cutting ants for odorant receptors (ORs), elongases, gustatory receptors (GRs), major royal jelly and yellow proteins (MRJP/y), and cuticular proteins (CPRs).**

|  | ORs | Elongases | GRs | MRJP/y | CPRs |
|---|---|---|---|---|---|
| *A. echinatior* | 456 (138) | 14 (2) | 62 (7) | 24 (2) | 32 (8) |
| *A. insinuator* | 424 (65) | 12 (3) | 61 (5) | 21 (3) | 31 (8) |
| *A. heyeri* | 467 (181) | 11 (4) | 61 (8) | 22 | 31 (9) |
| *A. charruanus* | 429 (72) | 12 (1) | 60 (9) | 22 (2) | 33 (9) |
| *P. argentina* | 263 (73) | 11 (3) | 27 (4) | 13 (4) | 31 (10) |
| *At. colombica* | 472 (72) | 14 | 62 (9) | 20 (1) | 34 (8) |
| *At. cephalotes* | 454 (114) | 13 (1) | 62 (5) | 17 (5) | 32 (7) |

Incomplete genes (due to misannotations, misassemblies, or pseudogenization) are given in parentheses.

compared to hosts (Supplementary Information). Gene losses in social parasite lineages occurred significantly more often in clades that had previously experienced gene gains during the radiation of free-living *Atta* and *Acromyrmex* leaf-cutting ant species (blue highlights in Fig. 4A, $X^2 = 15.14$, $p = 1e−04$, df = 1), suggesting that particularly leaf-cutting ant specific OR expansions were eroded by gene loss in the socially parasitic species in these lineages. Clade 285 of the 9-exon OR subfamily exhibited the most significant reductions, with free-living leaf-cutting ant species having 11–18 OR genes and social parasites having 1–10 paralogs, converging towards the 1–4 paralogs typically found in the phylogenetically more basal non-leaf-cutting fungus-growing ants that have much smaller, less complex and shorter-

lived colonies (Fig. 4B). Gene tree reconciliation uncovered an independent expansion of clade 285 in both *Acromyrmex* and *Atta* followed by recurrent gene losses only in the socially parasitic *Acromyrmex* species. Overall, the reconciled tree contained 44 duplication and 28 deletion events, of which only one duplication but 15 deletions were assigned to socially parasitic lineages, a highly significant difference ($X^2 = 24.18$, $p = 9e−07$, df = 1).

Hierarchical clustering analyses of OR subfamily sizes revealed a clear divergence between leaf-cutting species and their sister group, the non-leaf-cutting fungus-growing ants (Fig. 5A). While the less modified social parasites *A. insinuator* and *A. charruanus* clustered next to their hosts, the dramatically reduced OR

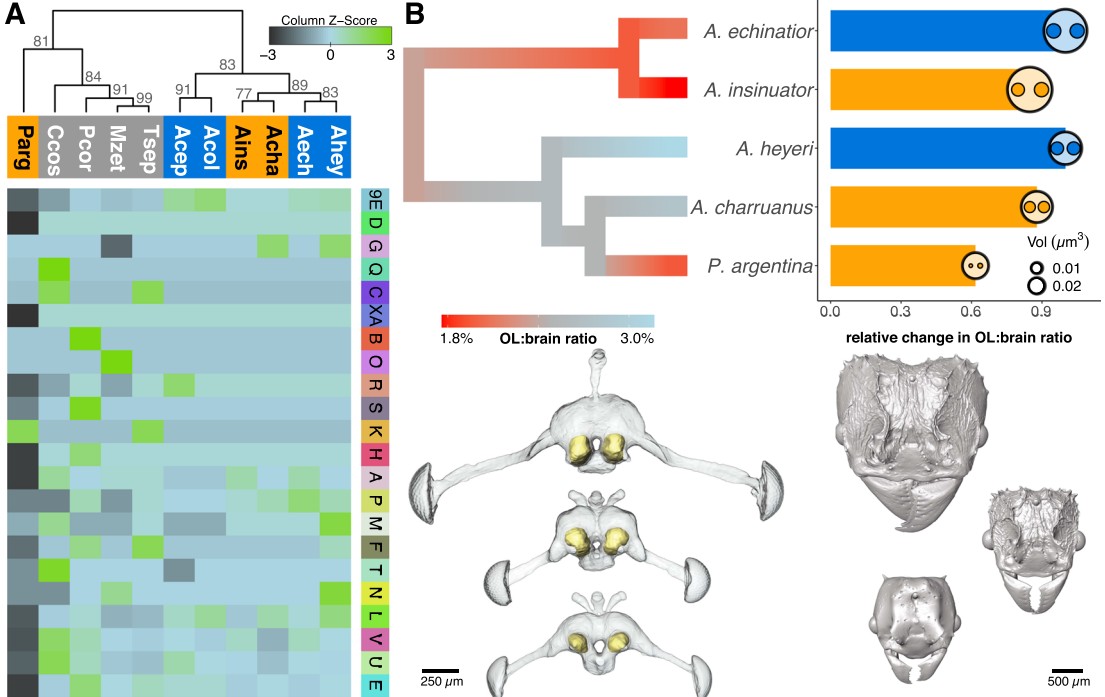

**Fig. 5 Regression of olfactory systems in inquiline parasites. A** Hierarchical clustering analysis of fungus-growing ant species based on OR subfamily sizes. Note that the social parasites *A. insinuator* (Ains) and *A. charruanus* (Acha) cluster according to their joint parasitic lifestyle and that *P. argentina* (Parg) clusters with sister lineages of non-leaf-cutting fungus-growing ant species. The heatmap is scaled by row to emphasize differences between species. The legend column towards the right corresponds to the outer gene-family ring in Fig. 4A. Numbers at the nodes of the dendrogram give approximately unbiased p-values (AU) from multiscale bootstrap resampling. **B** Relative olfactory lobe (OL) size of the hosts and inquilines. The phylogram is an ancestral state reconstruction of OL volumes relative to total brain volumes across the social parasites (*A. insinuator*, *A. charruanus* and *P. argentina*) and their hosts (*A. echinatior* and *A. heyeri*). Barplots show ratios of OL volume to total brain volume in inquiline parasites (in orange) relative to their hosts (in blue). Circles inserted at the tips of bars are proportional to the measured total brain volumes, while the smaller contained circles represent the measured volumes of the right and left OLs. On average, Panamanian species have larger brains than Uruguayan species (2-sample *t*-test, $p_{t-test} = 0.005$, df = 2.97, t = −7.74, n = 5). Relative OL volumes became reduced ($p_{t-test} = 0.059$, df = 2, t = −2.65, n = 5) as inquiline social parasites evolved their different degrees of specialization along the gradient of inquiline adaptations known as the inquiline syndrome[27]. Shown below are 3D surface reconstructions of the brains (with the OLs highlighted in yellow) and of the head capsules of *A. heyeri*, *A. charruanus,* and *P. argentina* (from top to bottom).

repertoire of *P. argentina* made this social parasite cluster away from the leaf-cutting ant species together with representatives of *Cyphomyrmex*, *Mycetomoellerius*, *Paratrachymyrmex*, and *Trachymyrmex* (until a recent revision[59], the latter three belonged to the paraphyletic grade previously referred to as *Trachymyrmex*). That the two less modified *Acromyrmex* social parasites clustered together based on their socially parasitic life history rather than their phylogenetic co-ancestry suggest that OR gene repertoires tend to become convergently reduced relatively early during the inquiline specialization process (Fig. 5A).

Previous studies have suggested that during ant evolution the expansions in size and glomeruli number of the olfactory lobes (OL) have remained correlated with the expansion of the OR gene repertoire[60,61]. Using microCT scans, we explored whether the relative size of the OL was consequently smaller in inquiline social parasite species compared to their hosts, mirroring the reduced OR repertoire encoded in their genomes. As hypothesized, relative OL size was ca. 12% smaller in *A. charruanus* and ca. 38% smaller in *P. argentina* when compared to their host *A. heyeri* (Fig. 5B), and analogously OL size was reduced by ca. 15% in *A. insinuator* relative to its host *A. echinatior*. This reduction in social parasites was marginally non-significant (one-sided *t*-test, $p = 0.059$, df = 2, t = −2.65). The results should be regarded as preliminary because only a single individual per species could be brain-imaged due to the rarity of sufficiently well-preserved samples of the social parasites. Notwithstanding this caveat, the preliminary analysis of brain morphology provided here suggests

that reductions are proportional to the degree of specialization found in social parasites.

We also explored whether gene losses in social parasites were apparent in other gene families than in the OR family. We produced detailed annotations of four candidate gene families: (1) gustatory receptors (GRs), which are involved in chemical communication similar to odorant receptors[62]; (2) Major Royal Jelly (MRJP)/yellow genes, which play a role in caste differentiation[63,64]; (3) elongases, which contribute to the production of cuticular hydrocarbons[65]; (4) and cuticular protein genes (CPRs), which are involved in exoskeleton formation[66] (Table 2). We detected substantial losses in the GR family in *P. argentina*, comparable to the extreme reduction of ORs in this species, with only 31 GRs being retained in the genome of *P. argentina* in comparison to 69 copies in its host *A. heyeri*. However, the number of GRs was not reduced in the other two social parasite species (*A. insinuator* 66 GRs, *A. charruanus* 69 GRs). Similarly, the MRJP/yellow family was reduced in *P. argentina* (17 genes), whereas all other analyzed species had between 20 and 26 copies (median = 23, Table 2). Our analyses of the elongases and CPR gene families did not reveal significant differences between any of the species (Table 2).

**Genome rearrangements.** To gain a more detailed understanding of the putative mutational events accountable for gene losses in social parasite genomes, we analyzed gene synteny and whole-

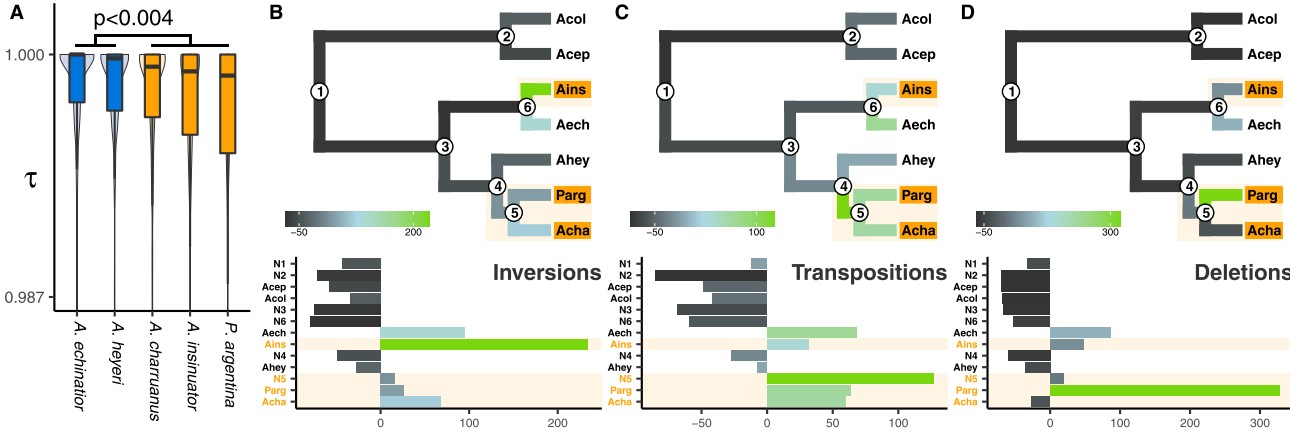

**Fig. 6 Microsyntenic changes and genomic rearrangements in socially parasitic and free-living leaf-cutting ant species. A** Microsyntenic changes relative to the *Atta colombica* outgroup are more frequent in social parasites (*A. charruanus*, *A. insinuator*, *P. argentina*) compared to their free-living host species (*A. echinatior*, *A. heyeri*). Syntenic conservation was quantified using Kendall's correlations of orthologous genes, with $\tau = 1$ for perfectly conserved syntenic regions and $\tau < 1$ for syntenic regions containing microsyntenic changes, and evaluated for significance with one-sided Mann–Whitney tests: hosts vs parasites: $p = 0.0037$, $z = 2.67$, $n = 610$; *A. echinatior* vs *A. insinuator*: $p = 0.0191$, $z = 2.06$, $n = 236$; *A. heyeri* vs *A. charruanus*: $p = 0.3223$, $z = 0.46$, $n = 250$; *A. heyeri* vs *P. argentina*: $p = 0.0091$, $z = 2.36$, $n = 246$. Boxplot centers show the median, hinges show the first and third quantiles, and whiskers show the 1.5 x inter-quantile range. **B–D** Social parasite genome evolution is characterized by an excess of inversions in *A. insinuator* (**B**), transpositions in the stemgroup of *A. charruanus* and *P. argentina* (**C**), and deletions in *P. argentina* (**D**). Compared to *A. insinuator* and *P. argentina*, the *A. charruanus* genome experienced relatively few deletions, transpositions and duplications after the split from its common ancestor with *P. argentina*, which was characterized by a very high number of transpositions.

genome alignments (Fig. 6). Using gene orthology information, we assessed microsynteny of the *Acromyrmex* and *P. argentina* genomes relative to *At. colombica* within larger syntenic blocks and quantified synteny conservation by calculating gene order correlation coefficients (see Supplementary Information). Two perfectly syntenic regions have a correlation coefficient of $\tau = 1$, whereas microsyntenic rearrangements, such as inversions and translocations, decrease the value for $\tau$. Comparing the three social parasite species with their two hosts species, $\tau$ was significantly decreased, indicating that microsynteny is less conserved in social parasites than across free-living *Acromyrmex* species ($p_{\text{Mann–Whitney}} = 0.004$, Fig. 6A). Analyses of mutation spectra using whole-genome alignments provided further evidence for faster synteny decay in social parasites as their genomes showed increased frequencies of inversions (Fig. 6B), transpositions (Fig. 6C), and deletions (Fig. 6D). Inversions were uniquely abundant in the *A. insinuator* genome, transpositions in the reconstructed genome of the ancestor of *P. argentina* and *A. charruanus*, and deletions were rampant in the genome of *P. argentina*. These analyses are broadly consistent with those plotted in Fig. 2B–E and suggest a higher fixation prevalence of genomic rearrangements followed by accelerated synteny losses in the genomes of social parasite lineages. This result is remarkable because the free-living attine ants already have significantly elevated rates of synteny loss, not only compared to eukaryotes in general but also relative to other ants[47].

## Discussion

Our comparative study revealed that the genomes of the three inquiline social parasite species *P. argentina*, *A. charruanus*, and *A. insinuator* have substantially diverged from their closely related *Acromyrmex* hosts. However, the details of the structural divergences differ, because *A. insinuator* evolved inquilinism independently from the other two social parasites that share a common ancestor but diverged and evolved independently for the past 1.63 million years. Our data indicate that these divergences from free-living hosts are reflected in genome-wide and gene-family-wide

signatures of relaxed selection, an increased abundance of various types of genomic rearrangements, and a substantial history of gene loss, contrasting previous studies[67]. Gene losses were detected most prominently in chemosensory gene families including olfactory and GRs and they accumulated over just 1–2.5 million years. Such dramatic evolutionary changes are only expected after fundamental changes in social niche and life history and are predicted to be at least partly convergent when such changes are analogous. Our results largely match this scenario because ancestral selection pressures on traits directly related to cooperative social colony life, such as chemosensory perception for communication, synaptic plasticity for behavioral differentiation, and secretion of semi-ochemicals for social interactions could all be alleviated because social parasites came to exploit the foraging efforts, nursing behavior, and colony infrastructure of their hosts.

The fundamental changes in life history experienced by ant inquiline social parasites are on par with those of intracellular endosymbionts and the most specialized non-social parasites, and they likely surpass transitions made by subterranean and cave-dwelling animals where a social or symbiotic niche dimension is lacking. The consistent signatures of relaxed selection coinciding with regressions and functional losses in phenotypic traits concur with other genomic studies that compared specialized lineages with their non-specialist sister clades that remained much closer to the ancestral lifestyle[14,20,54,68]. The small effective population sizes and low population densities of social parasites, as well as their frequent inbreeding via sib-mating, all reduce the efficacy of natural selection and increase the prevalence of genetic drift, simultaneously enhancing the rates by which ancestral traits are lost and constraining the rates by which novel adaptive traits can emerge. Similar mechanisms have been demonstrated to underlie globally relaxed selective constraints in obligate plant parasites[14] and intracellular endosymbionts[69,70], both of which experienced significant effects of drift and reduced gene flow due to their specialized life histories and modified population structures. Genome evolution in ant social parasites therefore strikingly resembles the dynamics known from other species that transitioned to highly specialized and unusual niches.

The 'unusual' here is important because the genomes of ants have evolved for more than 100 million years under a selection regime that emanated from a single major transition to a novel superorganismal level of social organizational mediated by a unique gene regulation network[71] for obligate queen-worker caste segregation based on unconditional altruism[72]. This produced extant biodiversity of 17 subfamilies, 337 genera, and more than 13,800 extant species[73]. It is therefore no surprise that parallel shifts to a socially parasitic lifestyle abandoning this fundamental ancestral condition, usually based on outbreeding and larger effective populations, leave significant genomic footprints[31]. The results of our analyses of just three of these inquiline species confirm that ant social parasites offer important 'loss-of-function' study systems for identifying hallmarks of cooperative social colony life at both the phenotypic and the genomic levels[31,67]. Analogous evolutionary dynamics have been identified in Myxozoa, microscopic obligate parasites that evolved from multicellular cnidarian ancestors, where many genes essential for complex multicellularity were secondarily lost when parasitic species regressed to a dramatically reduced body plan[16].

Ant genomes evolve relatively fast with rapid losses of synteny and quickly changing gene content[47,52,53], potentially related to the high recombination rates of the social Hymenoptera that transitioned to superorganismality[74–76]. A previous comparative genomics study of fungus-growing ants further demonstrated that rates of synteny loss are significantly higher in attine ants than in ants in general[47]. The fact that Acromyrmex inquilines show even faster rates of synteny loss underlines that their genomes might be prone to very rapid modification, which is consistent with ant inquiline lineages being almost invariably of recent origin[28]. The acceleration of evolutionary change under relaxation of selective constraints likely also contributed to the emergence of novel adaptive traits in social parasites[31], although at a slower rate because selection needs to be strong to overcome erosion by drift. Life as an inquiline social parasite universally requires new and highly specific adaptations to host finding, intrusion into mature host colonies, and manipulation of the hosts' brood care behavior to favor the non-related inquiline brood. Some of these adaptations have been studied in A. insinuator[77–79], A. ameliae[80], and a number of recent reviews have also underlined the consistency of convergent adaptations across ant inquiline social parasites[28,31,81]. The fact that we did not detect clear genomic signatures of adaptive change shared across the three independently evolved social parasite species may well be because convergent evolution does not necessarily rely on the same set of genes, particularly in lineages heavily affected by drift, or because changes were concentrated in regulatory, non-coding regions. It is also possible that adaptive changes are mediated epigenetically[31]. These aspects were beyond the scope of our present analyses and deserve further study.

We found that OR repertoires were consistently reduced across inquiline parasites, with the most dramatic reduction in the phenotypically and behaviorally most specialized inquiline P. argentina. These results need to be evaluated in light of the general expansion of the OR gene family as key evolutionary innovation associated with the emergence of complex social behavior in ants in general[55,60,61,82]. Inquiline ants rely on the brood care behavior and social infrastructure provided by their host colony and therefore depend to a lesser degree on the sophisticated and fine-tuned general chemical communication systems of their closely related free-living hosts, as long as they are adapted to successfully recognize and manipulate host colonies. Inquilines have in fact lost most ancestral social traits and do not engage in brood-care, colony founding, foraging, or nest-defense behaviors (e.g.[77,78]), so it is unsurprising that many ORs became obsolete and were consequently no longer maintained by

natural selection. The similarities of these losses to the regressive evolution of olfactory repertoires in blood-sucking parasites[17–19], the optic tectum in cave fish[83], and the erosion of plastomes in parasitic plants[14] are quite striking.

In contrast to ORs, we did not find convergent reductions in GRs, CPRs, MRJPs, and elongases, suggesting either that these gene families are maintained by natural selection in social parasites, or alternatively, that these genes are in close linkage to neighboring genes maintained by selection. OR genes are usually arranged in homogeneous and well-defined tandem arrays in the genome[55]. This increases the likelihood of non-allelic recombination due to their inherent repetitiveness[84], driving both diversification and reduction of gene families arranged like this. Such scenario could possibly explain both the general accumulation of OR diversification in free-living ants and the rapid secondary loss in the inquiline species studied here. Intriguingly, the only other gene family where we found substantial reduction in at least one social parasite was the GR family, which also has long tandem arrays in ant genomes.

The dramatic reduction of both ORs and GRs in P. argentina is associated with an extremely modified morphological and behavioral phenotype (Table 1, Fig. 5). It is intriguing that P. argentina is also highly divergent when compared to A. charruanus, which has not nearly the same extent of pronounced reduction in body size, OL, antennal segments, and internal mouthparts[41]. This implies that P. argentina and A. charruanus have evolved at very different rates, both at the genomic and phenotypic levels. Since the divergence of the two species, the rates of codon sequence evolution and gene loss decreased in A. charruanus, but increased further in P. argentina. In particular, the frequencies of genome rearrangements (transpositions) were increased in the common ancestor of A. charruanus and P. argentina and continued to increase in P. argentina, but was stalled during the later evolution of A. charruanus, so the overall rate of genome evolution came to resemble the rates observed in the free-living, non-parasitic Acromyrmex species. It is possible that A. charruanus quickly reached a stable fitness plateau after gene flow with ancestral Pseudoatta had ceased, or that new hybridization/introgression with its host somehow constrained further genome evolution, but any such considerations remain speculative with the currently available data. Pseudoatta argentina is the only social parasite species in our comparison where secondary host shifts are known because in addition to parasitizing A. heyeri, P. argentina has also been recorded to usurp colonies of A. lundii[41,43,58]. Consistent with secondary host acquisition, P. argentina has a significantly larger distribution range than A. insinuator and A. charruanus, inhabiting large parts of northern Argentina, Uruguay, and southern Brazil. However, just like the other inquilines, P. argentina is always exceedingly rare and has low population densities, rendering this species as difficult to study as almost all other ant inquilines[41].

Among natural historians there is no doubt that inquiline socially parasitic ants have both gained adaptive traits required to successfully invade, usurp and exploit their host colonies, as well as lost many ancestral traits associated with the free-living cooperative life style they abandoned[31,67]. For example, the phenotypically distinct worker caste is likely under strong stabilizing selection in all superorganismal lineages such as the ants, corbiculate bees, and vespine wasps[72]. Shifting from the cooperative social to an exploitative socially parasitic niche represents a major reductive shift in organizational complexity that has convergently taken place many times. It is important to note, however, that these reductions are not reversals to a solitary life history reminiscent of the ancestors of the ants in the early Cretaceous, because social parasites retain all dependencies on an integrated social colony context. Because inquiline populations

are also invariably much smaller than those of their hosts and inbreeding often documented[27,35], reduced effective population sizes have also been a natural assumption. Previous reviews have suggested that low effective population sizes of social parasites should, all else equal, reduce the evolutionary potential of inquiline social parasites[85,86]. However, for the key traits that determine inquiline reproductive success, all else is probably not equal because selection for exploitative traits will be strong once an incipient inquiline lineage becomes confronted with defenses that hosts can more easily evolve because their effective population sizes are larger. That the ant social parasite species studied here diverged considerably from their closely related free-living relatives, and that these differences evolved in a short period of time, supports the conclusion that the evolvability of inquiline social parasites remained high for traits that matter for the parasitic life history, even though inquiline genomes became marred by deletions, inversions, and transpositions. Considering that inquiline species evolved over 40 times independently across the ant tree of life[28], future genomics studies of these social parasites are likely to generate exciting further insights, particularly with long-read sequencing technologies allowing analyses in even greater detail. Such studies should address both the erosion of reproductive division of labor when selection for complementary and highly cooperative caste phenotypes is relaxed and, more generally, the signatures of convergent evolution for which few other model systems can offer species-level sample sizes of several dozens.

## Methods

Detailed methods and additional information is provided in the Supplementary Information.

**Sample collection, sequencing, assembly, and annotation.** Colonies of *Acromyrmex heyeri* and its social parasites *Acromyrmex charruanus* and *Pseudoatta argentina* were collected at Plantación Cruz Roja, Uruguay. *Acromyrmex insinuator* was collected from colonies of its host *Acromyrmex echinatior* in Gamboa, Panama. All work with live ants was conducted in compliance with ethical regulations for animal testing and research. DNA was isolated from several dozens of ants collected from a single colony for each species using a CTAB-based method, RNA was isolated from pools of 10–30 males for *A. insinuator*, or pupae, males, and queens, for *A. charruanus* and *P. argentina*, or workers for *A. heyeri* always using the RNeasy Plus Universal Mini Kit (Qiagen).

De novo genome assemblies for the genomes of *A. charruanus*, *A. heyeri*, *A. insinuator*, and *P. argentina* were produced from ~30 Gb (*A. insinuator*), ~34 Gb (*P. argentina*), ~35 Gb (*A. charruanus*), and ~40 Gb (*A. heyeri*) short-read sequencing data, using six different insert-size libraries (200, 500, and 800 bp paired-end libraries, and 2, 5, and 10 kb mate-pair libraries) for each genome. After filtering low-quality raw read data, reads were further corrected based on K-mer frequency. First, we indexed each read with 17-mer to build a library of 17-mer frequencies. Second, we corrected bases that were likely wrongly sequenced, based on the concept that bases wrongly sequenced typically yield low frequency k-mers. Initial genome assemblies were generated with SOAPdenovo on corrected reads that passed quality control. Gap filling was done for each assembly using paired-end reads of short insert sizes. To generate comparable gene annotations, we (re-)annotated all eleven attine genome analyzed in this study using the same annotation pipeline. For annotating protein-coding genes of the four de novo sequenced ant species (*A. charruanus*, *A. heyeri*, *A. insinuator*, and *P. argentina*) and seven previously published attine genomes (*A. echinatior*, *At. cephalotes*, *At. colombica*, *C. costatus*, *M. zeteki*, *Pa. cornetzi*, and *T. septentrionalis*), we generated homology-based, de novo and transcriptome-based predictions and combined the results of three methods in glean v1.0.1 to produce integrated gene sets for all species. Combined predictions from homology, de novo and transcriptome-based methods are summarized in Supplementary Table 21.

Whole protein sequences of 17 species (*A. echinatior*, *At. cephalotes*, *At. colombica*, *Camponotus floridanus*, *C. costatus*, *Dinoponera quadriceps*, *Harpegnathos saltator*, *Linepithema humile*, *Monomorium pharaonis*, *M. zeteki*, *Ooceraea biroi*, *Pa. cornetzi*, *Pogonomyrmex barbatus*, *Solenopsis invicta*, *T. septentrionalis*, *Vollenhovia emeryi*, and *Wasmannia auropunctata*; see Supplementary Table 22) were used as references to perform homology-based gene predictions. We used augustus v2.5.5 with default parameters to produce de novo predictions with homology-based annotations from *A. echinatior* as the training set. For transcriptome-based gene predictions, RNAseq data generated from pooled individuals (see above) were aligned to the genome of each corresponding species with Tophat v2.1.0 (options: -r 20 -mate-std-dev 10 -I 50000 -solexa1.3-quals).

Secondly, we used Cufflinks v2.0.2 (with -I 10000) to predict gene structures according to the alignment results from Tophat. We screened genomes for tandem repeats with Tandem Repeats Finder and for transposable elements (TEs) using a combination of LTR_FINDER, PILER, RepeatProteinMask, Wu-BlastX, and RepeatMasker.

**Phylogenetic analyses and divergence estimates.** We used orthofinder to establish orthology of protein-coding genes and to define a set of single-copy orthologs between all attine species included in our study. We aligned peptide sequences of single-copy orthologs with prank and back-translated to CDS with pal2nal. After removing orthologs showing evidence of recombination (with Phi) and orthologs with putatively saturated sites, we extracted fourfold degenerate sites using custom scripts (4dSites.pl available at https://github.com/schraderL/inquilineGenomics). Using a Maximum Parsimony-inferred tree as starting tree (see Supplementary Information), we reconstructed the phylogeny of the 11 species under a Maximum Likelihood framework with RaxML v8.2.12 and under a Bayesian framework using MrBayes v3.2.6 in unpartitioned analyses on the concatenated alignment of fourfold degenerate sites. We used MCMCtree (PAML v4.9 h) for inferring divergence date estimates between the different attine species and we calibrated the tree using fossil evidence and estimates of genus splits published previously (see Supplementary Information).

**Demographic inference.** We used MSMC2 v2.1.1 (available at https://github.com/stschiff)[87] to infer effective population sizes over time for *A. echinatior*, *A. octospinosus*, and *A. insinuator* from single individual sequencing data of two individuals per species collected in Gamboa, Panamá (Supplementary Table 33). We used short-read (100 bp), small-insert paired-end Illumina sequencing aiming to generate 20× coverage per individual. After quality control with trimmomatic, we mapped the qualified paired-end reads to the *A. echinatior* reference genome with bwa and removed duplicates with picard. We generated multihetstep and mask files for all scaffolds and each individual. We restricted our analysis to larger scaffolds over 2 MB and reduced time segments to $1*2 + 15*1 + 1*2$ to avoid overfitting, following recommendations by the developers of MSMC2[87]. We ran 500 bootstrap replicates for all four haplotypes per species.

**Evolutionary rate analyses.** Analyses of signatures of selection were performed on single-copy orthologs (SCOs) across the available leaf-cutting ant genomes (*Acromyrmex* and *Atta*). We retrieved 7750 single-copy orthogroups with orthofinder, aligned protein sequences with prank using the inferred species tree (see above) as guide tree, and generated CDS alignments with pal2nal. Orthogroups that showed significant signs of recombination ($p < 0.01$ as tested with Phi) were removed.

We identified genes showing significant evidence for positive selection by running adaptive branch-site tests (absREL, implemented in HYPHY v2.3.14) on CDS alignments and the species tree as input. Similarly, we tested for relaxed selection in protein-coding single-copy ortholog genes by running RELAX (implemented in HYPHY v2.3.14). RELAX fits three $d_N/d_S$ rate categories under the alternative model to the test and reference branch sets and infers a"selection intensity parameter" k, to test for relaxation or intensification along the specified test branches compared to the reference set. $K < 1$ is indicative of relaxed natural selection and $k > 1$ suggests intensification in the test compared to the reference set. Relaxed natural selection here refers to both relaxation of positive selection (i.e., $d_N/d_S$ average $> 1$ and $d_N/d_S$ test-set closer to 1 than $d_N/d_S$ reference-set) and to relaxation of purifying selection (i.e. $d_N/d_S$ average $< 1$ and $d_N/d_S$ test-set closer to 1 than $d_N/d_S$ reference-set). Conversely, intensification of positive selection is inferred if $d_N/d_S$ average $\geq 1$ and $d_N/d_S$ test-set farther from 1 than $d_N/d_S$ reference-set and intensification of positive selection is inferred if $d_N/d_S$ average $< 1$ and $d_N/d_S$ test-set farther from 1 than $d_N/d_S$ reference-set. The model is fitted so that the first two rate categories ($d_N/d_{S\ 1,2}$) summarize regions in the coding sequence with $d_N/d_S < 1$ (i.e. purifying selection, summarized in Fig. 2C) and the third category ($\omega_3$) summarizes regions with $d_N/d_S > 1$ (i.e. positive selection, summarized in Fig. 2D). We calculated background omega ratios $d_N/d_{S\ bg}$ from $d_N/d_{S\ 1}$ and $d_N/d_{S\ 2}$ for each gene from the test and reference branch sets ((as $d_N/d_{S\ bg}$ = (proportion$_1$ · $d_N/d_{S\ 1}$) + (proportion$_2$ · $d_N/d_{S\ 2}$)), and we tested whether these background evolutionary rates were significantly different between the test and reference sets. Similarly, we compared $d_N/d_{S\ 3}$ rates between both branch sets. For comparing average evolutionary rates across hosts and parasites, we removed genes with average $d_N/d_S > 10$ in at least one branch. This resulted in sets containing 3850 single-copy orthologs (i.e., 49.7 % of all 7750 single-copy orthologs) in the absREL analysis and 3616 (i.e., 46.7 %) in the RELAX analysis. GO enrichment analyses were performed with topGO on sets of genes showing intensification (with $k > 1$, FDR $< 0.1$) or relaxation (with $k < 1$, FDR $< 0.1$) of selection in the social parasite branches.

**Codon usage bias.** We calculated GC content at the first (GC1), second (GC2), and third (GC3) codon positions, as well as the effective number of codons (ENC) for each CDS in single-copy orthologs across species of *Acromyrmex*, their inquiline social parasites, and *Atta*. We then compared ENC~GC3 plots and neutrality plots to assess the extent of codon usage bias in the different species.

**Gene family size evolution.** We computed gene family clusters between all available genomes from *Acromyrmex* species and *At. colombica* using blastp all-vs-

all searches across predicted proteomes and subsequent MCL clustering (MCL v.14-137). We ran blastp v.2.6.0+ using an e-value cutoff of 1e−5. We excluded non-leaf-cutting attine species to reduce parameterization complexity of the modeling. Following gene family clustering, we excluded gene families that contained proteins identified as TE proteins or TE-derived proteins, based on annotations of all proteomes done with TransposonPSI v.08222010. Using CAFE v4.0, we modeled gene family size evolution in *Acromyrmex* host and social parasite genomes based on gene family clusters predicted with MCL. CAFE was run on the dataset comprising the ultrametric species divergence tree, the estimated species-specific error rates, and the gene family clustering. After fully parameterized models repeatedly failed to converge in CAFE, we ran all possible variations of 2-parameter models, to estimate branch-specific $\lambda$ and $\mu$ values. All 2-parameter models were run independently at least eight times to produce robust parameter estimates. The best-scoring $\lambda$ and $\mu$ parameter estimates were clustered by Hartigan-Wong $k$-means clustering ($k = 2$ to $k = 8$), choosing each time 1000 random sets as initial centers (nstart = 1000). The best fit to the data was a model with six discrete rates of $\lambda$ and $\mu$. To confirm that the extreme rates of gene loss inferred for *P. argentina* were not due to a possible incompleteness of gene annotations, we repeated the analysis of gene family sizes with independently generated homology-based gene annotations (produced with GeMoMa) for all included species (see Supplementary Information). GO enrichment analyses with topGO were run on gene families consistently smaller in the social parasites compared to their hosts and on gene families consistently smaller in the hosts compared to the social parasites. The latter analysis did not yield any significantly enriched GO terms. Similarly, no GO terms were enriched in lineage-specific genes in each of the three social parasites.

**Whole-genome alignment and synteny**. We generated whole-genome alignments for all attine species with the phylogeny-aware aligner progressiveCactus v0.1 (default settings), informed by the inferred species tree. We then used halSummarizeMutations to summarize inferred mutations at each branch of the underlying attine phylogeny. We calculated transposition (P), insertion (I), deletion (D), inversion (V), and duplication (U) events per million years of evolution, based on inferred divergence estimates from the phylogeny. We grouped mutation events by size (i.e., number of affected bases) for each branch in the phylogeny and calculated for each mutation type the relative number of events per Ma as the percent increase or decrease compared to the mean number of mutational events across all branches.

Synteny was inferred based on the conserved order of ortholog genes across *At. colombica* and the five *Acromyrmex* and *Pseudoatta* species using i-ADHoRe v3.0 using ortholog information between protein-coding genes from orthofinder v2.2.6. We processed the inferred pairwise multiplicons in R, restricting our analyses to longer regions spanning at least 25 genes in each genome. This filtering reduced the number of analyzed syntenic regions from 5425 to 1803. We calculated a Kendall's rank correlation coefficient ($\tau$) as a quantifier for syntenic conservation for each syntenic region (see details in Supplementary Information), with perfectly syntenic regions having a coefficient of 1. We tested for a significant decrease in syntenic conservation relative to *A. colombica* comparing the two host genomes (*A. echinatior* and *A. heyeri*) to the three inquiline social parasite species (*A. insinuator*, *A. charruanus*, and *P. argentina*) using a one-sided approximate (Monte Carlo) Wilcoxon–Mann–Whitney Test.

**Odorant receptor annotation**. ORs were annotated in all available attine genomes. We initially retrieved manually curated OR annotations for *A. echinatior*, *At. cephalotes*, and *Solenopsis invicta* published by[61]. Gene annotations for *A. echinatior* were again manually refined using unpublished *A. echinatior* antennal RNAseq data (pers. comm. Bitao Qiu). Most of these manual refinements were adding short first exons that were intentionally left out in the original curated gene set (McKenzie pers. comm.). We used these manually curated annotations to manually annotate OR genes in *A. insinuator*.

Based on the manual curated gene sets from *A. echinatior*, *A. insinuator*, and *At. cephalotes*, we then annotated OR genes in genomes using a gene annotation pipeline that combines evidence from the homology-based gene predictors GeMoMa and exonerate as well as blast hits with EvidenceModeler (see Supplementary Information). For each species, the final set of predicted ORs comprised models coding for an intact 7tm6 domain (predicted with pfam_scan) and models with significant similarity to known ORs (using blastp runs against the set of manually curated ORs). We reconstructed the OR gene phylogeny for eleven attine species using amino acid translations of all predicted genes. First, we assigned proteins to subfamilies by blasting protein sequences against a gene family assignment database (Sean McKenzie pers. comm.) built from previously reconstructed ant OR phylogenies[82]. We assigned ORs to subfamilies according to the best blastp hit against the assignment database and created subfamily-wide protein alignments using linsi v7.307. Subfamily alignments were then merged using mafft v7.307 and the phylogenetic tree was computed using FastTreeMP v2.1.10. OR clades were defined using the R function getCladesofSize() (clade.size = 5) from the phytools library. Gene trees were reconciled as follows: We retrieved CDS sequences for all genes of a given clade with GenomeTools' (v1.5.9) extractfeat and trimmed CDS sequences to multiples of three while maintaining the ORF with TransDecoder.LongOrfs. The retained CDS sequences were aligned with prank

v.150803 in codon mode and a phylogeny inferred with FastTree -nt. For the detailed analysis of clade 285, we rooted the gene tree at OR CcosOr-363 before running dlcpar in search mode and inferring tree-relations with dlcoal v1.0.

**X-ray microtomography**. Ethanol-preserved samples of *P. argentina*, *A. insinuator*, *A. charruanus*, *A.heyeri* and *A. echinatior* were used for X-ray Micro-tomography (microCT) to visualize three-dimensional brain morphology and quantify OL volume. Samples were stained in Lugol's solution (2.5% potassium iodide, 2.5% iodine) for 2–8 days and destained in 70% ethanol for 2–6 h. Specimens were scanned wet in 70% ethanol, mounted inside a polyimide tube sealed on one side with hot glue and on the other with mounting putty. MicroCT was performed with a General Electric phoenix v-tome-x m x-ray computed tomography scanner (GE Sensing Inspection Technologies, Wunstorf, Germany), equipped with a 180 kv nanoCT tube at the Smithsonian Institution National Museum of Natural History. Scanning parameters are presented in Supplementary Table 50. Reconstruction of 3D images was completed with GE datos-x v 2.4.0.1199 and resulted in a stack of DICOM images with voxel sizes of 1.99–2.41 m. Segmentation and visualization of ant specimens were performed in Amira (FEI v5.5.0) and brain and OL volumes were calculated using the MaterialStatistics module.

**Reporting summary**. Further information on research design is available in the Nature Research Reporting Summary linked to this article.

## Data availability
Raw sequencing data, genome assemblies, and annotations have been deposited in GenBank with BioProject accession codes PRJNA552756, PRJNA605929, and PRJNA714946 and in the CNGB Sequence Archive with accession codes CNP0000530 and CNP0000905. Further data are available in figshare with identifier https://doi.org/10.6084/m9.figshare.14224253. Previously published ant genomes were retrieved from GenBank.

## Code availability
Scripts and code used for data generation and analyses are archived on Github (https://github.com/schraderL/inquilineGenomics, https://doi.org/10.5281/zenodo.4608504)[88] and documented in the Supplementary Information.

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

## Acknowledgements

This study was supported by grants from the European Research Council (ERC Advanced grant 323085) (J.J.B.), the US National Science Foundation (DEB-1654829 and CAREER DEB-1943626), the University of Rochester, and Arizona State University (C.R.), as well as the Lundbeck Foundation (R190-2014-2827) (G.Z.). The Visitor's Office of the Smithsonian Tropical Research Institute provided logistic help and facilities to work in Gamboa and the Autoridad Nacional del Ambiente y el Mar (ANAM) gave permission to sample and export ants from Panama. We gratefully acknowledge the Dirección General de Recursos Naturales Renovables for permission to conduct field research in Uruguay, and Daniel Ramírez from Cambium Forestal for facilitating access to our Uruguayan field research sites. M.B. thanks Leticia Tejera for her discovery of the *P. argentina* population.

## Author contributions

J.J.B., C.R. and G.Z. (in alphabetic order) conceived of the study, M.B., C.R., J.J.B. and M.S. collected and extracted the samples, H.P., X.B. and Y.D. performed sequencing, assembly, and annotation, L.S. performed the phylogenomic, demographic and evolutionary rate analyses, analyzed gene family size evolution, micro synteny, and genome rearrangements. F.J.L. performed the microCT analyses. L.S. designed the figures. L.S., J.J.B. and C.R. wrote the manuscript with contributions from G.Z. and H.P. All authors read and approved the final manuscript.

## Competing interests

The authors declare no competing interests.
