## [Peer Review File · Nature Communications]

REVIEWER COMMENTS

Reviewer #1 (Remarks to the Author):

The authors sequenced and performed extensive comparative genomic and phylogenetic analyses on several ant species to get a better understanding of the evolutionary genetics of social parasitism in fungus-growing ants. They tell a self-consistent and clear story of relaxed selection / genetic drift associated with the small population size and no-longer-required worker ant traits. In particular, they follow up on the evolution of the olfactory receptors (smaller gene families in the parasitic species) including careful manual curation and documenting a potentially associated putative phenotypic change, namely smaller olfactory lobes.

This paper clearly represents an extensive investment of time, expertise, and money; the story as told is both interesting and valuable. Unfortunately, there are some methodological issues that may or may not have a major influence on the overall results, and that need to be resolved before the story can even be properly evaluated.

Major issues:

1. Genome quality -

As I'm sure the authors are all too aware by now, Illumina-only genomes are just about antiquated. The cost and effort required with current long read or other long-distance information (e.g. Hi-C, 10-x) sequencing technologies to make a major improvement on the presented assemblies is likely very minor compared to the existing investment. While I appreciate no one wants to go back to this step near publication, some of the presented analyses (coalescent analyses, synteny analyses, and honestly also gene annotation) are directly tied to genome quality and contiguity. For instance, the authors report that only scaffolds above 2Mbp were included for coalescent analyses, with an N50 of only 1.15Mbp for *A. insinuator* (Tab S5), they must have been working with substantially less than half of their data. While the other three species have a contig N50 around an order of magnitude shorter than current long read technologies consistently produce for read N50.

Basically, if the biological material is available for even

a modest amount of modern sequencing this could be a fairly easy avenue to improve the genome quality and the base-data for the entire study.

As much as I think extra sequencing would be worth it, and I would encourage it if it is a possibility, I understand that constraints of material and time might mean this is simply not possible.

2. Annotation quality (particularly *P. argentina*) -
This is the single most important point in this review.

I was positively surprised to read that the authors went through the effort of re-annotating 7 existing genomes on top of the 4 they sequenced to have comparable annotation quality.

Not everyone would have done this, and it's appreciated.

Unfortunately, while the authors could guarantee the same annotation pipeline, this pipeline does not appear to have produced the desired comparable annotation quality for all species. The complete BUSCO scores for the annotations range between 96.0% and 89.1% (Supplementary section 8). A lot of the ensuing paper reports gene loss in the species with 89.1% of BUSCO's in its annotation (*P. argentina*). The authors repeatedly discuss gene loss in *P. argentina* as a biological result. However, it stands out that an impressive 97.1% of the same BUSCOs could be identified in the genome of *P. argentina*. That makes the 89.1% look a lot like a technical issue in the annotation and not a biological result.

This is not even addressed in the text as an issue.

For the olfactory receptor gene family, the authors describe independent and careful curation of the annotations, and they confirm, for this family, gene loss in *P. argentina*.

Because the low annotation BUSCO score was not explicitly addressed (or please tell me I missed it), I don't know if the authors meant to hold up the OR family as evidence this resulted from real genome-wide gene loss. But if they did, there are a couple reasons this is not appropriate / sufficient.

- it's generally insufficient to hold up one piece of evidence against a large genome-wide pattern.
- it completely ignores and fails to explain the high BUSCO score of the genome.
- it's possible that some still unidentified biases that affected the main annotation pipeline could also have affected the secondary annotation/curation.

For instance, was the same RNAseq data used (also see major point 4)? Or was the glean annotation used as a starting point for the other?

So basically, I strongly suspect that the BUSCO drop (genome 97.1% - annotation 89.1%), is a gene annotation issue and that the 89.1% is not related to gene loss, and that the 'signature' of gene loss in *P. argentina* that flows through the paper is an artifact. Luckily, story wise I don't think it's a deal breaker, but it needs to be cleaned up, and it needs to be reported honestly.

I would first suggest that the authors redo (some of) the annotations, and aim not for identical pipelines, but for comparable results. Personally, I would suggest measuring the annotation quality not by the raw BUSCO scores but by the `_change_` in BUSCO scores from genome to annotation. A small increase in complete BUSCOs is normally expected for a good annotation. I would encourage the authors to tweak the pipeline as necessary to achieve comparable results. If they have to bring in an extra homology species because homology mapping works less effectively for one of the 11 targets, then do it. If they have to change the coverage threshold of cufflinks because they have less or different-quality RNAseq data for a species, then do it.

Obviously, annotation quality won't be identical, but if the difference remains substantial (large enough to influence any biological interpretations) then this needs to be reported and addressed, by:

- if possible, investigate and discuss why annotation isn't working properly
- annotation-independent confirmation of gene loss (e.g. blast back to the genome and make sure to include all raw contigs w/o any pre-filtering. Honestly, this is a good idea anyways).
- address the expected repercussions of technical issues in the text.

3. Statistical and data interpretation check -

I have substantial concerns about some of the statistical and data analysis choices as currently presented. The good news is that this is going to be far easier to address than the issues above.

Example A)

The authors report no multiple hypothesis correction for any of the GO-enrichment analysis. From the perspective of the functional enrichment, it is less important `_how_` some gene family set was defined (e.g. whether the significance threshold for positive or negative selection was $FDR < 0.05$, $FDR < 0.1$, or $p < 0.001$) and more important

that multiple hypothesis correction is performed on the enrichment results themselves. A side effect of not doing this can be seen in all the small GO terms (e.g. Table S40, Annotated ≤ 13 , Significant = 1) that appear enriched because a single gene family is in the set. This is stochastic, a lot of noise, and is expected in almost any smallish gene family set chosen. At the very least or better yet in combination with multiple hypothesis correction, pre-filtering tiny annotation terms would help.

Example B)

The authors report, plot and interpret extremely high dN/dS values of up to 1000. As a sanity check/thought experiment on this, I would ask the authors to consider what they think the probability of truly having 1000 non-synonymous changes to a single synonymous change would be. Remember, you can't select against the synonymous change. After that I would encourage them to look at the actual alignments behind dN/dS values of 1, 5, 10, 100, and 100. My guess is that the very high values are miss-alignments and the inflation calculations meant to correct for multiple substitutions per site explode in unfortunate ways. Alternatively maybe the very high values are stochastic e.g. $dN=1/dS=0$ returning a value that is high but defined because a little epsilon got added somewhere. After these exercises (no need to report or include them, but the authors should do them for their own understanding), I would ask the authors to choose another much-lower-than-1000 and more reasonable threshold. I realize this might not seem like it should be the authors job to figure out, and maybe it shouldn't be, but if they are using bioinformatics in their paper, I fear using tools cautiously and skeptically is part of the job. :-)

Example C)

line 260 the authors report a 'marginally significant' (one-sided t-test $p < 0.059$). The difference could have been in either direction. It would have been more surprising, but still interesting in the other and this really isn't an appropriate place for a one-sided t-test. In fact, with just one sample per species and so few species there's very little statistical power and I don't think it's an appropriate place for a statistical test at all. This just increases the potential confusion due to the unfortunate yet exceedingly common mix up of 'no significant difference' and 'no difference'. The raw data and the explanation that authors anyways include on the following line are better on their own than with iffy statistics giving an illusion of legitimate confirmation to some while having the opposite effect on the statistically inclined.

Example D)

And this is just a question as I really don't know / I'm not enough of

a phylogenetics effort to critique this.

Is it appropriate to use different binning of the years in the coalescent analysis (Fig 2A) and then compare between species?

I do not have the time nor to the ability (e.g. Example D above) to do a full critique of data analysis, presentation and statistics on my own. I would recommend the authors do a careful review and double-check of their own work and in all cases of doubt, please consult an expert. I expect that the end result of this will largely be focusing the paper on more substantial and meaningful differences, with less distraction and confusion.

4. Method communication -

While it's an easy and understandable mistake to make, many parts of the methods and more technical results are insufficiently described, and this definitely needs to be fixed.

Example E)

When thinking about issue 2 above, I tried to get some feel for the RNAseq data and how it was used.

- 1st, I feel like the authors could be a bit more descriptive than 'various castes and life stages', in particular, were there any potential differences between species? I imagine at the very least that the same castes were not available for the parasitic species, but I don't want to imagine, I want to simply read what was done.
- 2nd, were there statistics on the RNAseq data anywhere? Total read counts or number of genes predicted by cufflinks, etc...?
- 3rd, was the RNAseq data used in GeMoMa or otherwise in the manual curation of OR annotations. Your reader should be able to understand what sort of data is going in and out without an intimate familiarity with the GeMoMa command line and certainly the reader shouldn't have to know what the output files from cufflinks were named.

Example F)

When the authors say they performed a BUSCO analysis on the annotation, is this a BUSCO in mode 'prot' (proteins) or 'tran' (transcripts).

Example G)

Line 290, what was the statistical test and what was the resulting test statistic, how did the test treat the data from the different species? Also, were parasitic species simply pooled and treated as if the values therein were independent, or were the species accounted for?

Example H)

I couldn't find listed which 'different insert-size libraries' were simple paired end reads, and whether the larger sizes were perhaps mate pair libraries? Yet the size distribution would indicate something of the sort must be the case.

Please don't get me wrong here, the authors obviously put a lot into reporting and documenting their methods and technical results, better than many for sure. Still, there were some major oversights, which need to be fixed, of which the above is a sample and does not attempt to be a complete list. I encourage the authors again to double check things with fresh eyes after having had to wait on us reviewers, and they will probably catch more than I.

Minor issues

Overall the paper was an extremely smooth and easy read and I have only a few things here.

Line 398, just thinking: Assuming the gene loss in the OR family holds up to the requested additional scrutiny: I'd be curious if it were a series of deletions or have you lost one or few tandem blocks thereby losing many genes at once.

Line 323, I don't quite follow what data or numbers the authors are referring to when they claim that the changes in inquiline ants are 'on par' to those in intracellular endosymbionts. That is quite the claim without numbers.

Line 125, I think the word 'site' is missing after 'degenerate'

Line 155, I believe the authors meant to write $dN/dS < 1$ for relaxed purifying selection

Is the incomplete link in the Supplemental table 29 header supposed to be there? If so, how should the reader access this?

Log spelled differently (lg, log2) in two subplots within Sfig 27

Overlapping labels Sfig 28, 30 (offsets or similar could help).

Figure 2B&C - this might be easier / more interpretable if combined

into one, (particularly after addressing issue 3, example B), might also not, but I'd encourage the authors to try and see if it makes sense while simplifying the plot.

Figure 2E - the inset plot is sufficiently similar in zoom to the wider original it doesn't really show us anything not already apparent in the larger plot.

Every figure using the viridis color scale. I personally would recommend describing the low color as purple, not blue. It's a bit nit picky but the scale goes through blues, the extreme is the purple.

Finally

Thanks, I learned somethings reading the paper, and despite the major issues that need addressing, it is a substantial and interesting work already.

- Dr. Alisandra K Denton

Reviewer #2 (Remarks to the Author):

Review of Schrader et al. (Nature Comm.)

Overall, this manuscript was a very nice read. The analyses and interpretations are good. My criticisms of the manuscript are about not what is presented, but rather what is not presented (omissions). There are omissions of the relevant literature, there are omissions of analyses that were done, and there are analyses that are not done. These omissions leave the impression that there is a bias in the framing of the manuscript – this is unlikely the intent of the authors, but a byproduct of the publication process (perhaps an over-focus on the story at the expense of the data). At any rate, these omissions are addressable. Not addressing these makes the manuscript both less valuable, but also makes it incomplete and biased.

The largest omissions from the literature are missing citations (either completely lacking or lacking at the appropriate places). There are published studies, some remarkably similar in basic premise to this one, that are not cited. Why? The contrast, for example, between the results of genomic changes in *Pogonomyrmex/Vollenhovia* social parasites relative to their hosts (Smith et al. 2015, *Mol. Biol. Evol.*) with the results of this study are interesting. There are interesting differences in methods and also differences in results – these differences are enriching rather than distracting to the story that is being told. Omitting mention of these other results is very problematic. There are other, similar papers, that could also be cited much earlier in the manuscript, for example the Cini et al. perspective/theory paper on social parasite evolution. This paper is referenced in passing much later in the manuscript. Another omission referenced in one of my later comments is a paper by Feldmeyer et al. (2017) in *BMC Evol. Biol.* – this looks at some genomic changes between slave-making parasites and their hosts.

The omissions of results are also troubling. The storyline is intensely focused on the changes in the social parasites. This is understandable, but by only reporting on what has changed in the parasites ignores changes in the hosts. For example, families that have high gene losses in social parasites are discussed, but did the hosts also lose genes? By ignoring this symmetry (bifurcation between host/parasite) we lose half the story and bias both the story and the conclusions. For example, what processes (GO terms) are enriched in the genes/families lost by the hosts? Looking at these is essentially a control for looking at what happens in the parasites.

The omissions of analyses that were not done, or are not reported, are, for example, analysis of lineage-specific genes or analysis of regulatory elements. I do not intend to say that these need to be included, but a large focus on just what is lost (i.e., gene loss) and not what might be gained (novel genes) is a bias. This bias could be addressed in text – contextualize the results. It is ok to say that the analysis was not done, but to not admit that only looking for losses will bias the conclusion toward finding losses is a slight to the reader.

Below are specific comments.

L105-107: It seems a falsehood, at best, to term this manuscript the first to explore molecular evolution of social parasitism. This phrasing does not honor previous studies (some going back decades). Perhaps the authors are relying on the word ‘detailed’ in this sentence, but that is too subjective a word. This is not even the first genomic study on social parasitism. This should just be deleted.

L125: you are missing the word “sites”?

L155: Do you mean $dN/dS < 1$ for purifying?

L161: I am not sure the results suggest “erosion” of social parasite genomes. Yes, the results (as presented) suggest that protein coding genes are under less intense selection, relative to non-parasites. So, relative to non-parasites the protein coding portion of the genomes of social parasites is under more relaxed selection --- this is not ‘erosion’. Insufficient information is given to come to the conclusion of ‘erosion’ of the protein coding genome. Erosion makes me think that the genome is ‘falling apart’, but rather it is changing. Additionally, it would be useful to report sample size of genes in each comparison (including as a percent of those annotated or the shared ortholog set used in the analysis) – that is, how many genes in parasite and host lineages have $dN/dS > 1$, etc.? What is the percent of those genes of the total in the dN/dS analysis?

L167: Report this also as a percent. Presumably it is out of the ortholog set which is 6338, so 4%? This is a lot, I think. Same applies to the next part on intensified selection.

L179: I am not convinced that there is evidence of ‘massive genome erosion’ (as previously stated). Furthermore, what genes are under relaxed/intensified selection in (at least one of) the hosts and what

are their enriched GO terms? There is a focus on what has changed in the social parasites and nothing about change in the hosts. Without equal attention to both parasite and host it is hard to tell whether the observed changes are unique to the parasites and whether the GO terms make much sense.

L186: An interesting analysis that is almost conspicuously missing is 'novel' gene gains. The ortholog set is relatively small (half of the annotations), suggesting that there are a number of lineage specific genes in each of the lineages. These analyses only look at gains/losses in gene families, but this is not likely the whole story. I am not saying that this type of analysis is essential to include, but if it has been done then it should be included. And if it is not yet done then perhaps acknowledging what is unknown is relevant – that is, contextualize your results to say that there are gene losses detected in your analysis, but that your analysis is not comprehensive. By only focusing on what you did find, and ignoring what you didn't (or didn't look for) biases your narrative.

L187: 'an' outgroup

L201: What other families experienced losses? Omissions like this, and the lack of comparative data from hosts as noted above, are useful for story-telling, but do not tell an unbiased story. I continue to get the impression that the authors are more focused on telling a story with a predefined end than putting the data 'on the table'. Here I think that the contraction in ORs is very interesting, but were there other contractions that could also be interesting? Are there contractions that are consistent in the hosts? My guess is probably not, but this is not clearly articulated in the manuscript and so doubts are sown by omissions. ***Note, as I read down in the ms I see the other gene family analyses. I decided to leave in the previous comment because I think it is relevant to address the order of analyses in the manuscript. I suggest that a reference to the other family analyses appears in the first sentence of the OR gene family analysis. Or, a modified version of this whole paragraph goes to the beginning of the section where you begin gene family analysis.

L351: Similar statements are made by other papers, but are ignored here (see Feldmeyer et al. 2017, BMC Evol. Biol., Smith et al. 2015 Mol. Biol. Evol.). Furthermore, the perspective/theory paper by Cini et al. that is cited is not cited here. These sorts of omissions reflect poorly on this paper.

L441: I am not too sure about precedent for this phrase, but I would recommend a change from 'evolutionary potential' to 'selective potential' to simply reflect that the predominant evolutionary mechanism in question is selection (i.e., a decrease in the potential for selection, relative to drift, when N_e is low).

L447: This language could be in the Smith et al. (2015) paper that looked at genomic change between social parasites and hosts, too. Again, the omission of comparing the results of these papers is a major omission.

Reviewer #3 (Remarks to the Author):

In this paper the authors investigate whether the evolution of parasitism is associated with a decrease of the strength of selection on some gene families. The study reveals some interesting pattern despite the fact that there are only 3 parasitic species (and 2 events of evolution of parasitism). The authors characterize global patterns of relaxed constraint, and then move on to focus on the large olfactory receptor family, noting large reductions. microCT were also used to ask if olfactory lobe has changed size in a correlated way. The authors end the paper on a quantification of different mutational classes (inversions, transpositions, deletions), showing largely consistent results – increased mutational events along parasite branches (with some exceptions).

Overall, the results of the paper are quite interesting and of general interest, and the data seem of high quality. We do, however, have concerns/questions regarding a number of the analyses.

- 1) An important issue is whether the patterns uncovered are due to changes in social behaviour per se or whether it is due to a decrease of N_e (which is expected to occur with the evolution of social life). Surprisingly, the authors assess changes of N_e for only 2 species. It would be useful to do this for all the studied species (and also compare the data with other published work -see Romiguier et al. JEB 2014.).
- 2) The authors discuss various "selection" analyses throughout the paper but need to state more precisely the type of selection the test are investigating. For example, it is not clear what is meant w/ "relaxed positive selection" (Fig2D) using dNdS values. Similarly, on L175, "intensified selection" could mean several things (but positive selection is what the authors are testing for), and on L218 "relaxed selection" should be "relaxed purifying selection".
- 3) Could the authors also investigate codon usage bias as a way of detecting relaxed purifying selection? In addition, linkage disequilibrium is expected to be increased in bottlenecked populations, so perhaps this would be an additional line of evidence, and another interesting analysis to contrast with estimates of N_e .
- 4) For inference of N_e using 2 genomes, it would be helpful to have a citation of the method in the sentence (L145). See also study of Romiguier et al. mentioned above.
- 5) The authors do not discuss the results of the MSMC2 analyses much beyond the smaller N_e in *A. ins* compared to *A. ech*. What do the authors make of the inferred recent increase in N_e , that puts the value

above the initial (though uncertain) starting N_e ? This should be extended with data from the other species.

6) Fig 2C and 2D: These two panels are not clear. Is panel E just a continuation of panel C, and if so, why are there few data points for values just smaller than 1 and many data points for values just greater than 1.

7) Fig2E: Not sure much is gained by the inset, but adding a carpet where data points lie would be helpful. It would also be helpful for readers unfamiliar with the RELAX parameters being estimated to have a simpler way of displaying the meaning of x-axis of Fig2E.

8) L159: It would be helpful to have numerical summaries of genes that fall into the "minority" and "majority" categories.

9) The interpretation from the GO analyses are not particularly convincing (here and in the discussion). These categories are broad enough to construct many stories.

10) The authors did not find accelerated gene loss in *A. char*, but do not really discuss this much. Why would gene loss be stopped along only this lineage? This is where an N_e estimate for this genome might be helpful. (see comment 15 also)

11) The results on OR losses are interesting. More details/quantifications about number of full/large deletions, nonsense mutations etc. would be interesting. It has been shown in *Drosophila* that there is some type of 'readthrough' of OR nonsense mutations (and maybe other genes), and thus that pseudogenes can't be annotated alone by this (<https://www.nature.com/articles/nature19824>). The mechanism remains unclear, so it may/may not persist across more distant taxa.

12) L208 and L251: Not sure that that ORs losses should be called "convergent" unless these were independent losses of orthologs. It would be more appropriate to use a term such as "parallel losses of OR family members", or something along these lines.

13) One concern with the OR work is that the authors study this gene family (and a few more more such as GRs, MRJP,CPR)in more careful detail than other families. It would be useful to have a balanced gene family-wide analysis. Though it seems like ORs are real outliers and present very interesting evolutionary dynamics, the special treatment of this family does leave open the possibility for some selection bias.

14) The way the hypothesis on L253 is started off is a bit misleading: number of glomeruli does not necessarily have to relate to overall OL size. The count of specific glomeruli is a different than the volumetric measure obtained with the microCT scans (at least as I understand it). Also it would probably be more honest to call the result from the these scans as marginally "non-significant" (L260).

15) For the genome rearrangements section, do the authors have any way to gauge the relative

confidence in the annotation across mutational classes? Are deletions harder to identify, for example than inversions? *D. cha* stands out again as having a paucity of deletions. In that respect, having details about the mutational events removing OR could be helpful. Also, It would seem possible that the lack of deletions in *D. cha* is related to the relative lack of gene loss (as shown in Fig3A). If not mistaken, this seems worth discussing.

16) Regarding the rearrangements, how many genes are involved/associated in these events?

17) L311: not sure that it was really gene-family specific. There were multiple gene families in the GO list (table S40), and ORs were on the top but this does not mean it was specific – just enriched.

18) An idea: might it make more sense to keep the “Genome rearrangements” section together with the section “geneomic evidence of relaxed selection”? It seems that these data got together, and it reads a bit odd to return to mutation types after going through the olfactory sections.

18) L413: deletions do not look to be increased in common ancestor, right?

19) L125: typo in: “.4-fold degenerate inferred...”

Roman Arguello and Laurent Keller

Reviewer #1 (Remarks to the Author):

REVIEWER COMMENT:

The authors sequenced and performed extensive comparative genomic and phylogenetic analyses on several ant species to get a better understanding of the evolutionary genetics of social parasitism in fungus-growing ants. They tell a self-consistent and clear story of relaxed selection / genetic drift associated with the small population size and no-longer-required worker ant traits. In particular, they follow up on the evolution of the olfactory receptors (smaller gene families in the parasitic species) including careful manual curation and documenting a potentially associated putative phenotypic change, namely smaller olfactory lobes.

This paper clearly represents an extensive investment of time, expertise, and money; the story as told is both interesting and valuable. Unfortunately, there are some methodological issues that may or may not have a major influence on the overall results, and that need to be resolved before the story can even be properly evaluated.

Major issues:

1. Genome quality -

As I'm sure the authors are all too aware by now, Illumina-only genomes are just about antiquated. The cost and effort required with current long read or other long-distance information (e.g. Hi-C, 10-x) sequencing technologies to make a major improvement on the presented assemblies is likely very minor compared to the existing investment. While I appreciate no one wants to go back to this step near publication, some of the presented analyses (coalescent analyses, synteny analyses, and honestly also gene annotation) are directly tied to genome quality and contiguity. For instance, the authors report that only scaffolds above 2Mbp were included for coalescent analyses, with an N50 of only 1.15Mbp for *A. insinuator* (Tab S5), they must have been working with substantially less than half of their data. While the other three species have a contig N50 around an order of magnitude shorter than current long read technologies consistently produce for read N50.

Basically, if the biological material is available for even a modest amount of modern sequencing this could be a fairly easy avenue to improve the genome quality and the base-data for the entire study.

As much as I think extra sequencing would be worth it, and I would encourage it if it is a possibility, I understand that constraints of material and time might mean this is simply not possible.

RESPONSE:

We agree with the reviewer that long-read sequencing technologies would likely yield improved assemblies compared to the ones we present in our manuscript. Unfortunately, we cannot generate additional sequencing data, because two of the social parasite species (*A. charruanus* and *P. argentina*) are so rare and elusive that we have so far not succeeded to collect more material. In addition, the Covid pandemic has rendered fieldwork to search for social parasites virtually impossible at the moment.

Over the last decade, we have spent many weeks in the field to study and collect the *Acromyrmex* social parasites in Latin America at extensive efforts and many fruitless collecting trips. Collecting these species involves finding the host species, digging up the entire fungus garden, and then screening the nest for social parasites among the hundreds or thousands of host workers. Finding, excavating and screening a single host colony can take up to a full day, rendering this a very low throughput task.

While *A. insinuator* could be collected with some reliability in colonies of *A. echinator* in Panama over the last decades, collecting *A. charruanus* and *P. argentina* in South America is still largely a matter of chance. Despite our continuous efforts over the last years, there was only a single occasion to collect enough individuals for genome sequencing for both of these species. We decided to go with Illumina paired-end sequencing, which was

the most reliable technique at the time. Given the priceless nature of our samples, we did not want to take any chances with sequencing.

To avoid biases by comparing Illumina-based with PacBio-based assemblies, we subsequently decided to stick to Illumina sequencing for all species analyzed in our study, which allowed us to also include previously published attine genomes (Nygaard et al. 2016).

We are convinced that the current assemblies are of sufficient quality to support the analyses we conducted and the conclusions we draw, in particular because we do not see an explicit bias in assembly quality between the three social parasite species and the hosts. However, we have added a sentence in the concluding paragraph of our discussion to emphasize that third generation sequencing technologies will likely offer valuable novel insights that we might not have been able to address in our analyses.

We now write:

“... future genomics studies of these social parasites are likely to generate exciting further insights, particularly with long-read sequencing technologies allowing analyses in even greater detail.”

Nygaard, S., et al. (2016). "Reciprocal genomic evolution in the ant-fungus agricultural symbiosis." Nat Comm 7(1): 12233.

REVIEWER COMMENT:

2. Annotation quality (particularly *P. argentina*) - This is the single most important point in this review.

I was positively surprised to read that the authors went through the effort of re-annotating 7 existing genomes on top of the 4 they sequenced to have comparable annotation quality. Not everyone would have done this, and it's appreciated. Unfortunately, while the authors could guarantee the same annotation pipeline, this pipeline does not appear to have produced the desired comparable annotation quality for all species. The complete BUSCO scores for the annotations range between 96.0% and 89.1% (Supplementary section 8). A lot of the ensuing paper reports gene loss in the species with 89.1% of BUSCO's in its annotation (*P. argentina*). The authors repeatedly discuss gene loss in *P. argentina* as a biological result. However, it stands out that an impressive 97.1% of the same BUSCOs could be identified in the `_genome_` of *P. argentina*. That makes the 89.1% look a lot like a technical issue in the annotation and not a biological result.

This is not even addressed in the text as an issue.

RESPONSE:

We are aware of the differences in BUSCO assembly and annotation scores for the different genomes and we agree with the reviewer that this is an important point to address and that we should have done that in the previous version. We now mention these differences in the first paragraph of the results.

We further agree that the differences in annotation efficiency could in principle bias the results of our gene family size evolution analysis, in particular since we find the highest rates of gene loss in *P. argentina*. Other analyses are, however: 1. Based on only single-copy orthologs present in all species (e.g. analyses of evolutionary rates), 2. Independent of gene annotations (e.g. the analysis of genomic rearrangements), 3. Independent of gene losses/gains (e.g. the analysis of synteny), or 4. Based on gene annotations produced independent of the general gene annotation pipeline (analyses of ORs, GRs, etc.).

With regard to the gene family evolution analysis, we would like to point out that CAFÉ, the software used for the analysis, includes a method to correct for incomplete assemblies or annotations. This is achieved by computing species-specific error rates, which are then included in the final modelling of gene family size evolution (see page 17 of the CAFÉ tutorial at <https://iu.app.box.com/v/cafetutorial-pdf>). We have included these steps in our analysis and wrote accordingly on page 31 of the Supplementary:

“CAFE allows to account for assembly and annotation errors of the analyzed genomes, by iteratively estimating error rates in the input data sets based on likelihood score optimizations under varying error models. We ran error estimation several times in order to identify global optima in the likelihood landscape. “

As expected from the higher discrepancy in BUSCO genome and protein scores for *P. argentina*, the strongest error correction was applied to this species with 0.129. Error correction for all other species was lower than 0.055 (see page 32 of the initially submitted Supplementary Material for details). We agree with the reviewer that this information is crucial to the interpretation of the gene family size analysis and we accordingly have added a sentence in the main text to emphasize that the gene family size evolution analysis corrects for potential incompleteness issues in the underlying gene annotations.

We now write:

“To explore gene family evolution in social parasites, we modelled gene family size changes with CAFE, accounting for potential biases from incomplete assemblies and annotations. “

Regardless of this, we have decided to completely repeat the analysis of gene family size evolution to make sure the patterns we find are independent of annotation quality. For this, we have produced entirely new gene annotations for all genomes involved in this analysis.

We annotated protein-coding genes in each genome using published gene annotation for ten ants belonging to the same subfamily as our study species (Myrmicinae). The annotations were produced using a strictly homology-based approach with GeMoMa.

Final BUSCO scores for these protein annotations against Hymenoptera odb9 were as follows:

ACHA: C:97.8%	[S:97.3%,D:0.5%],F:1.0%,M:1.2%,n:4415
AECH: C:97.5%	[S:97.0%,D:0.5%],F:1.2%,M:1.3%,n:4415
AHEY: C:95.8%	[S:95.3%,D:0.5%],F:2.4%,M:1.8%,n:4415
PARG: C:97.2%	[S:96.8%,D:0.4%],F:1.4%,M:1.4%,n:4415
AINS: C:97.8%	[S:97.3%,D:0.5%],F:1.0%,M:1.2%,n:4415
ACOL: C:97.5%	[S:97.3%,D:0.2%],F:1.2%,M:1.3%,n:4415

Based on these gene annotations, we repeated our analyses with CAFÉ, and obtained lower error estimates:

PARG: 0.022
ACOL: 0.000
AINS: 0.008
AHEY: 0.010
ACHA: 0.007
AECH: 0.014

The repeated analysis of gene birth (λ) and gene death rates (μ) with CAFE using these annotations confirmed the key findings we present in the manuscript, that:

- Rates of gene loss are highest in *P. argentina* ($\mu = 0.0025$, $\lambda = 0.0003$)
- Gene loss is more frequent than gene gain in the ACHA/PARG stem group ($\mu = 0.0010$, $\lambda = 0.0003$)
- *A. charruanus* is characterized largely by gene gains. ($\mu = 0.0002$, $\lambda = 0.0017$)

The only discrepancy between this novel and our original analysis is that in the new analysis *A. insinator* is modeled to have a higher rate of gene gains ($\lambda=0.0013$). Note however that estimates for λ and μ for *A. insinator* vary in both analyses (see Figure 24 and newly added Figure 28 in the updated Supplementary Materials), possibly due to the recent split from *A. echinator* followed by rapid evolutionary changes.

We have added this new analysis in the Supplementary Material (pages 44-46 of the revised Supplementary Material) summarizing the repetition of our analyses. We also refer to it in the method section of the manuscript and have published the code at <https://github.com/schraderL/inquilineGenomics>.

REVIEWER COMMENT:

For the olfactory receptor gene family, the authors describe independent and careful curation of the annotations, and they confirm, for this family, gene loss in *P. argentina*.

Because the low annotation BUSCO score was not explicitly addressed (or please tell me I missed it), I don't know if the authors meant to hold up the OR family as evidence this resulted from real genome-wide gene loss. But if they did, there are a couple reasons this is not appropriate / sufficient.

- it's generally insufficient to hold up one piece of evidence against a large genome-wide pattern.
- it completely ignores and fails to explain the high BUSCO score of the genome.
- it's possible that some still unidentified biases that affected the main annotation pipeline could also have affected the secondary annotation/curation. For instance, was the same RNAseq data used (also see major point 4)? Or was the glean annotation used as a starting point for the other?

RESPONSE:

As pointed out by the reviewer, the annotation of odorant receptor genes is indeed entirely independent of the general gene predictions. It is thus unlikely that a bias underlying the lower BUSCO scores generated for the general gene annotation applies to the annotation of ORs as well, in particular given that the repetition of the gene annotation using GeMoMa (which is at the core of the OR annotation pipeline) did not show any appreciable bias in BUSCO scores (see Response above regarding the gene family size evolution analysis).

The annotations of odorant receptor genes were done without RNAseq data, as ORs are almost exclusively expressed in the antennae, making it challenging to produce good RNAseq support even for readily available species. Our OR annotation pipeline uses manually curated ORs from three ant species as input and we put substantial effort into quality control of the finally produced gene models for each species. We also compared performance of our annotation pipeline with a pipeline specifically developed for OR annotation in ants by McKenzie & Kronauer (2018) and could confirm that both pipelines produced essentially identical results (there were only minor differences in detecting the first very short exon in some genes).

McKenzie, S. K. and D. J. C. Kronauer (2018). "The genomic architecture and molecular evolution of ant odorant receptors." *Genome Res* **28**(11): 1757-1765.

REVIEWER COMMENT:

So basically, I strongly suspect that the BUSCO drop (genome 97.1% - annotation 89.1%), is a gene annotation issue and that the 89.1% is not related to gene loss, and that the 'signature' of gene loss in *P. argentina* that flows through the paper is an artifact. Luckily, story wise I don't think it's a deal breaker, but it needs to be cleaned up, and it needs to be reported honestly.

RESPONSE:

We agree with the reviewer that the drop in BUSCO scores is likely a consequence of the gene annotation pipeline we used. However, we hope that the arguments provided above and the repetition of the gene family size analysis can persuade the reviewer that the pattern of gene loss is not an artefact of the lower BUSCO scores.

As outlined above, we are convinced that any effects related to incomplete gene annotations are either accounted for or did not affect our analyses (for reasons provided above).

We think that the strictly homology-based GeMoMa annotations produced for the repetition of our CAFÉ analysis are a useful comparative tool but not as thorough as the gene annotations we have in the manuscript, because the latter are based on a well-tested and full pipeline including e.g. ab initio gene models. While BUSCO scores are a good indicator of general quality, we do not think they should be the sole quantifier for annotation quality.

Regardless, we do share the reviewer's concern that the BUSCO score drop needs to be addressed in the main text of the manuscript. As also mentioned above, we have therefore added a sentence to the results section.

We now write:

"BUSCO scores for protein annotations ("protein" mode) were lower than for the genome assemblies, particularly for P. argentina (89.1% vs. 97.1%), but these differences could be accounted for in our analyzes of gene family size evolution (see below)."

REVIEWER COMMENT:

I would first suggest that the authors redo (some of) the annotations, and aim not for identical pipelines, but for comparable results. Personally, I would suggest measuring the annotation quality not by the raw BUSCO scores but by the `_change_` in BUSCO scores from genome to annotation. A small increase in complete BUSCOs is normally expected for a good annotation. I would encourage the authors to tweak the pipeline as necessary to achieve comparable results. If they have to bring in an extra homology species because homology mapping works less effectively for one of the 11 targets, then do it. If they have to change the coverage threshold of cufflinks because they have less or different-quality RNAseq data for a species, then do it.

RESPONSE:

We hope that the confirmation of the key results of our gene family size evolution analysis with new annotations are sufficiently persuasive to alleviate the reviewer's impression that our results would not be reproducible (particularly those in regard to *P. argentina*, where BUSCO scores differed most between assembly and protein set).

REVIEWER COMMENT:

Obviously, annotation quality won't be identical, but if the difference remains substantial (large enough to influence any biological interpretations) then this needs to be reported and addressed, by:

- if possible, investigate and discuss why annotation isn't working properly
- annotation-independent confirmation of gene loss (e.g. blast back to the genome and make sure to include all raw contigs w/o any pre-filtering. Honestly, this is a good idea anyways).
- address the expected repercussions of technical issues in the text.

RESPONSE:

We have now added a section in the Supplementary Material (page 13) to discuss different factors that could negatively affect BUSCO scores of our annotation pipeline, including technical and biological/evolutionary explanations for a reduction of BUSCO scores in the social parasite (rapid changes in coding sequence, rapid restructuring of gene structure and synteny). We would further like to point out that the results from the whole genome alignment and the analysis of ORs (and similarly the microCT analysis of the olfactory lobes) all confirmed that deletions are most abundant in *P. argentina*.

Following the Reviewer's suggestion, we have now also included an annotation-independent confirmation of gene loss by blasting back OR protein sequences against the raw genome assemblies for *A. charruanus*, *A. heyeri*, *A. insinator*, and *P. argentina* as well as the published assembly of *A. echinator*. The results are consistent with a reduction of OR repertoires in the social parasites. We have included this approach in the Supplementary Material (page 50).

REVIEWER COMMENT:

3. Statistical and data interpretation check -
I have substantial concerns about some of the statistical and data analysis choices as currently presented. The good news is that this is going to be far easier to address than the issues above.

Example A)

The authors report no multiple hypothesis correction for any of the GO-enrichment analysis. From the perspective of the functional enrichment, it is less important `_how_` some gene family set was defined (e.g. whether the significance threshold for positive or negative selection was $FDR < 0.05$, $FDR < 0.1$, or $p < 0.001$) and more important that multiple hypothesis correction is performed on the enrichment results themselves. A side effect of not doing this can be seen in all the small GO terms (e.g. Table S40, Annotated ≤ 13 , Significant = 1) that appear enriched because a single gene family is in the set. This is stochastic, a lot of noise, and is expected in almost any smallish gene family set chosen. At the very least or better yet in combination with multiple hypothesis correction, pre-filtering tiny annotation terms would help.

RESPONSE:

We agree with the reviewer that caution is needed when interpreting p-values of uncorrected Gene Ontology term enrichment tests. Not correcting for multiple testing is however the recommendations from the authors of the enrichment analysis tool that we applied (topGO), in particular when applying the parent-child tests that account for GO-term relationships (Grossmann et al. 2007).

We have included a short excerpt from the topGO documentation to outline the reasoning (page 19, version April 27, 2020) (Alexa, Adrian, and Jörg Rahnenführer. "Gene set enrichment analysis with topGO." *Bioconductor Improv* 27):

6.2 The adjustment of p-values

The p-values return by the getSigGroups function are raw p-values. There is no multiple testing correction applied to them, unless the test statistic directly incorporate such a correction. Of course, the researcher can perform an adjustment of the p-values if he considers it is important for the analysis. The reason for not automatically correcting for multiple testing are:

- *In many cases the raw p-values returned by an enrichment analysis are not that extreme. A FDR/FWER adjustment procedure can in this case produce very conservative p-values and declare no, or very few, terms as significant. This is not necessary a bad thing, but it can happen that there are interesting GO terms which didn't make it over the cutoff but they are omitted and thus valuable information lost. In this case the researcher might be interested in the ranking of the GO terms even though no top term is significant at a specify FDR level.*
- *One should keep in mind that an enrichment analysis consist of many steps and there are many assumptions done before applying, for example, Fisher's exact test on a set of GO terms. Performing a multiple testing procedure accounting only on the number of GO terms is far from being enough to control the error rate.*
- *For the methods that account for the GO topology like elim and weight, the problem of multiple testing is even more complicated. Here one computes the p-value of a GO term conditioned on the neighbouring terms. The tests are therefore not independent and the multiple testing theory does not directly apply. We like to interpret the p-values returned by these methods as corrected or not affected by multiple testing.*

In our manuscript, we predominantly used results from GO enrichment analyses for hypothesis building and subsequently follow up on the results (as done for the odorant receptor analysis). Similarly, we processed results of GO enrichment analyses further with REVIGO (Supek et al 2011), thus grouping related terms into "semantically similar" categories.

We have referred to GO enrichment test results in two paragraphs of the manuscript. In response to the criticism raised here by Reviewer 1 and also criticism raised by Reviewer 3, we have edited these paragraphs as follows:

We previously referred to the results of GO enrichment analyses in lines 167 to 179:

"We identified 233 genes showing evidence of relaxed selection in at least one of the social parasite branches (at $FDR < 0.1$ and relaxation parameter $k < 1$) among single-copy orthologs. Several GO terms enriched among these 233 genes were associated with secretion, lipid metabolic processes, neuron projection guidance, and insulin receptor signaling (see Supplementary Figure 14-16, Supplementary Table 31, GO enrichment analyses with parent-child < 0.05). Many of these genetic pathways may relate to caste differentiation (insulin signaling), chemical communication (secretion), and behavioral complexity (neuron projection guidance), but more detailed studies will be required to test these hypotheses. Conversely, we also detected signatures of intensified selection (at $FDR < 0.1$, $k > 1$) in 102 genes. Here, enriched GO terms were associated with e.g. TOR signaling and metabolic functions (Supplementary Figure 17-19, Supplementary Table 32, $p < 0.05$), suggesting that the convergent niche shifts to social parasitism also triggered adaptive changes within an overall context of more massive genomic erosion."

We have changed this paragraph and removed explicitly mentioning the results of the uncorrected GO enrichment analyses, while still referencing the Figures and Tables in the Supplementary Material.

The paragraph now reads:

"We identified 233 genes (3.34% of the tested genes) showing evidence of relaxed selection in at least one of the social parasite branches (at $FDR < 0.1$ and relaxation parameter $k < 1$) among single-copy orthologs (see Supplementary Figures 14-16, Supplementary Table 34). However, we also detected signatures of intensified selection (at $FDR < 0.1$, $k > 1$) in 102 genes (1.46%, Supplementary Figures 17-19, Supplementary Table 35),

suggesting that the convergent niche shifts to social parasitism also triggered adaptive changes within an overall context of more genomic erosion.“

We have also now added a sentence to the Supplementary Material that the GO enrichment analyses referred to in this paragraph showed no significantly enriched GO terms after correcting for multiple testing (at $FDR < 0.05$).

Apart from the paragraph cited above, we previously referred to a GO enrichment test in lines 201 ff of the manuscript:

“This showed that gene loss had most significantly affected genes with olfactory receptor (OR) functions (GO:0004984 “olfactory receptor activity”, $p < 3.2e-4$, Supplementary Table 40).”

We have now repeated also this GO enrichment analysis. After FDR correction, only GO terms associated with ORs remained significant.

We have accordingly changed the sentence and now refer to the results of the corrected GO enrichment analysis.

We now write in the main text:

“This showed that gene loss had most significantly affected olfactory receptors (ORs), as only GO terms related to olfaction were significantly enriched (e.g. GO:0004984 “olfactory receptor activity”, $FDR < 5.6e-6$, Supplementary Table 45).”

Supek F, Bošnjak M, Škunca N, Šmuc T (2011) REVIGO Summarizes and Visualizes Long Lists of Gene Ontology Terms. PLOS ONE 6(7): e21800. <https://doi.org/10.1371/journal.pone.0021800>

Grossmann S, Bauer S, Robinson PN, Vingron M. Improved detection of overrepresentation of Gene-Ontology annotations with parent child analysis. Bioinformatics. 2007 Nov 15;23(22):3024-31. doi: 10.1093/bioinformatics/btm440. Epub 2007 Sep 11. PMID: 17848398.

REVIEWER COMMENT:

Example B)

The authors report, plot and interpret extremely high dN/dS values of up to 1000. As a sanity check/thought experiment on this, I would ask the authors to consider what they think the probability of truly having 1000 non-synonymous changes to a single synonymous change would be. Remember, you can't select against the synonymous change. After that I would encourage them to look at the actual alignments behind dN/dS values of 1, 5, 10, 100, and 100. My guess is that the very high values are miss-alignments and the inflation calculations meant to correct for multiple substitutions per site explode in unfortunate ways. Alternatively maybe the very high values are stochastic e.g. $dN=1/dS=0$ returning a value that is high but defined because a little epsilon got added somewhere. After these exercises (no need to report or include them, but the authors should do them for their own understanding),

I would ask the authors to choose another much-lower-than-1000 and more reasonable threshold.

I realize this might not seem like it should be the authors job to figure out, and maybe it shouldn't be, but if they are using bioinformatics in their paper, I fear using tools cautiously and skeptically is part of the job. :-)

RESPONSE:

We fully agree with the reviewer that such extremely high dN/dS rates are difficult to interpret biologically, as it is unlikely that in a pairwise comparison of two sequences a stretch of coding sequence has 1000 non-synonymous changes over one synonymous change.

We would like to point out that the method we applied (RELAX) divides codon alignments into three rate categories of which one (rate category 3) summarizes over “fast evolving sites” with dN/dS rates of above 1. Thus, the high dN/dS rates are never averages over entire genes in this analysis.

As assumed by the reviewer, we consider such high values in many cases to be the consequence of $dN=1$ and $dS=0$ in at least one rate category of at least one of the branches. (In fact, the authors of HyPhy also suggest considering $dN/dS > 50$ as infinite, see <https://github.com/veg/hyphy/issues/1110#issuecomment-604411123>)

Following the reviewer's recommendation, we have visually inspected several of the alignments producing very high dN/dS estimates and could not find a systematic bias caused by misalignments that would affect our

interpretation of the results. Regardless, we agree with the reviewer that it is almost impossible to exclude that misalignments also contribute to at least some of these high estimates. The impact of such misalignments should however be equal in all branches and will thus not have introduced a systematic bias towards either the social parasites or the non-parasitic species.

To make sure that extreme dN/dS values are not driving the results of our analyses, we have now restricted the analyses to genes where dN/dS < 10. The updated analysis confirmed the findings from our initial analysis. We have adopted this updated analysis in the manuscript now and changed the sections in the main text and in the supplementary materials accordingly (page 27) to reflect this restriction to dN/dS <10.

REVIEWER COMMENT:

Example C)

line 260 the authors report a 'marginally significant' (one-sided t-test $p < 0.059$). The difference could have been in either direction. It would have been more surprising, but still interesting in the other and this really isn't an appropriate place for a one-sided t-test. In fact, with just one sample per species and so few species there's very little statistical power and I don't think it's an appropriate place for a statistical test at all. This just increases the potential confusion due to the unfortunate yet exceedingly common mix up of 'no significant difference' and 'no difference'.

The raw data and the explanation that authors anyways include on the following line are better on their own than with iffy statistics giving an illusion of legitimate confirmation to some while having the opposite effect on the statistically inclined.

RESPONSE:

Here, we disagree with the reviewer that this is not a place for a one-sided test. The test was applied to explicitly test the hypothesis that derived from our analysis of the odorant receptor repertoires, i.e. that olfactory lobes are smaller in social parasites. We however agree with the reviewer that applying a statistical test to such a small dataset is borderline.

Thus, we have rephrased the section and now report the test statistic as marginally non-significant (as also suggested by Reviewer 3):

"This reduction in social parasites was marginally non-significant (one-sided t-test $p < 0.059$, $df=2$, $t=-2.65$)."

REVIEWER COMMENT:

Example D)

And this is just a question as I really don't know / I'm not enough of a phylogenetics effort to critique this.

Is it appropriate to use different binning of the years in the coalescent analysis (Fig 2A) and then compare between species?

RESPONSE:

In Schiffels & Wang 2020, the authors of MSMC2 outline how the time patterning should be designed to avoid overfitting the data. By default, MSMC2 (which was designed for human genomes) uses 32 time segments with 28 free parameters. We have reduced this to 17 time segments ($1^*2+15^*1+1^*2$) in our analysis, following recommendations from the developers of MSMC2:

"We recommend to experiment with these settings, in particular when non-human data is analyzed, where sometimes the default settings in MSMC and MSMC2 are not appropriate because the genomes are substantially shorter and hence fewer parameters should be estimated." (in Schiffels & Wang, 2020, page 165)

(Schiffels S, Wang K. MSMC and MSMC2: The Multiple Sequentially Markovian Coalescent. *Methods Mol Biol.* 2020;2090:147-166. doi: 10.1007/978-1-0716-0199-0_7. PMID: 31975167.)

We have now added this information to the Supplementary Material (page 25).

REVIEWER COMMENT:

I do not have the time nor to the ability (e.g. Example D above) to do a full critique of data analysis, presentation and statistics on my own. I would recommend the authors do a careful review and double-check of their own work and in all cases of doubt, please consult an expert. I expect that the end result of this will largely be focusing the paper on more substantial and meaningful differences, with less distraction and confusion.

RESPONSE:

We have been very careful with designing our analyses and have reviewed, double-checked and refined our methods extensively over the course of the project. Just to give some examples, we have performed reannotations of all genomes included in our comparisons (as acknowledged by the reviewer), we modelled evolutionary rates with two different methods (absREL and RELAX), we tested and consequently excluded genes showing evidence for homologous recombination in the phylogenetic analyses and the analysis of signatures of selection, we have removed putatively saturated sites in the phylogenetic inference, we used both Bayesian and Maximum Likelihood methods for estimating phylogenies, we have included a “skew 2 normals” prior distribution in divergence estimates to account for incongruent previously published estimates of divergence of *Atta*, we have removed any gene families putatively including TE-encoded genes from the gene family size analysis). We furthermore provide very detailed and thorough documentation in the Supplementary Material (as noticed by the reviewer) and have published the complete code that we used for our analyses. We thus feel that we have been sufficiently rigorous with our data analysis and hope that the additional information provided in the revision can resolve the reviewer’s concerns.

REVIEWER COMMENT:

4. Method communication -

While it's an easy and understandable mistake to make, many parts of the methods and more technical results are insufficiently described, and this definitely needs to be fixed.

RESPONSE:

We have expanded every section of the methods in the main text to cover substantially more details of our various analyses. We now also include a statement at the beginning of the Method section that the Supplementary Materials provide more details and additional information on all methods and analyses performed.

REVIEWER COMMENT:

Example E)

When thinking about issue 2 above, I tried to get some feel for the RNAseq data and how it was used.

- 1st, I feel like the authors could be a bit more descriptive than 'various castes and life stages', in particular, were there any potential differences between species? I imagine at the very least that the same castes were not available for the parasitic species, but I don't want to imagine, I want to simply read what was done.

- 2nd, were there statistics on the RNAseq data anywhere? Total read counts or number of genes predicted by cufflinks, etc...?

RESPONSE:

We apologize for the lack of detail in describing the samples used for RNAseq and fully agree that it is necessary information. We have expanded the Supplementary Material (section 3) and method section accordingly. We now also include an overview of the RNAseq data in Supplementary Figure 1.

REVIEWER COMMENT:

- 3rd, was the RNAseq data used in GeMoMa or otherwise in the manual curation of OR annotations. Your reader should be able to understand what sort of data is going in and out without an intimate familiarity with the GeMoMa command line and certainly the reader shouldn't have to know what the output files from cufflinks were named.

RESPONSE:

We are not sure we understand the point raised here regarding the naming of cufflink files. As mentioned in a response above, the odorant receptor annotation was not supported by RNAseq data. Gene models that were used as reference (see McKenzie & Kronauer 2018) for the reannotation of ORs in our study were however informed by RNAseq support.

We have now changed section 17 in the Supplementary Material and the method section to avoid any confusion.

We now write on page 48 of the Supplementary Material:

“GeMoMa is a homology-based gene prediction program that uses amino acid sequence and intron position conservation of a reference genome (using a gff and fasta file of the reference genome) to annotate protein-coding genes in a target genome.”

We furthermore expanded the sentence in the method section of the manuscript to:

“... a gene annotation pipeline that combines evidence from the homology-based gene predictors GeMoMa and exonerate as well as blast hits with EvidenceModeler (see Supplementary Material).”

We hope this resolves the issues raised by the reviewer.

McKenzie, S. K. and D. J. C. Kronauer (2018). "The genomic architecture and molecular evolution of ant odorant receptors." *Genome Res* **28**(11): 1757-1765.

REVIEWER COMMENT:

Example F)

When the authors say they performed a BUSCO analysis on the annotation, is this a BUSCO in mode 'prot' (proteins) or 'tran' (transcripts).

RESPONSE:

We have added the information that we used the “prot” mode in the main text and the Supplementary Material.

REVIEWER COMMENT:

Example G)

Line 290, what was the statistical test and what was the resulting test statistic, how did the test treat the data from the different species? Also, were parasitic species simply pooled and treated as if the values therein were independent, or were the species accounted for?

RESPONSE:

We thank the reviewer for pointing this out. While the information was already included in the Supplementary Material (see page 61/62 of the Supplementary Material of the initial submission), we forgot to include it in the manuscript.

We used a Mann-Whitney Test, comparing the three socially parasitic species against the two non-parasitic species. Thus, species were indeed pooled.

Comparing host-parasite pairs individually, the results are as follows:

AECH vs AINS: test statistic = 2.0573, p-value = 0.01914

AHEY vs ACHA: test statistic = 0.4632, p-value = 0.3223

AHEY vs PARG: test statistic = 2.3559, p-value = 0.00907

We have now included the missing information in the Supplementary Material (page 75-76) and the main text as well as the Figure legend.

REVIEWER COMMENT:

Example H)

I couldn't find listed which 'different insert-size libraries' were simple paired end reads, and whether the larger sizes were perhaps mate pair libraries? Yet the size distribution would indicate something of the sort must be the case.

RESPONSE:

We have added this information to the main text. The sentence in the Method section now reads:

“De-novo genome assemblies for the genomes of A. charruanus, A. heyeri, A. insinator, and P. argentina were produced from ~30 Gb (A. insinator), ~34 Gb (P. argentina), ~35 Gb (A. charruanus), and ~40 Gb (A. heyeri) short-read sequencing data, using six different insert-size libraries (200 bp, 500 bp and 800 bp paired-end libraries, and 2 kb, 5 kb and 10 kb mate-pair libraries) for each genome.”

REVIEWER COMMENT:

Please don't get me wrong here, the authors obviously put a lot into reporting and documenting their methods and technical results,

better than many for sure. Still, there were some major oversights, which need to be fixed, of which the above is a sample and does not attempt to be a complete list. I encourage the authors again to double check things with fresh eyes after having had to wait on us reviewers, and they will probably catch more than I.

RESPONSE:

We appreciate the thorough review provided by the reviewer and are grateful for acknowledging the effort we put into reporting and documenting. We have carefully gone over all of the analyses included in the study to confirm that all analyses have been thoroughly designed and conducted. We feel that readers should now have the possibility to critically evaluate all our analyses, if not directly from the text then by using our extensive Supplementary Material and code collection.

REVIEWER COMMENT:

Minor issues

Overall the paper was an extremely smooth and easy read and I have only a few things here.

RESPONSE:

Thank you!

REVIEWER COMMENT:

Line 398, just thinking: Assuming the gene loss in the OR family holds up to the requested additional scrutiny: I'd be curious if it were a series of deletions or have you lost one or few tandem blocks thereby losing many genes at once.

RESPONSE:

To explore the mode of gene loss, we annotated OR gene arrays as stretches in the genome containing three or more OR genes separated by less than 30 kb. We found that *Pseudoatta argentina* had the fewest of such arrays (15). For the other social parasites, we found 19 arrays in Acha and 21 in Ains. The two hosts had 27 (Ahey) and 29 (Aech) arrays. Average gene counts per array were 8 (Aech, Ahey), 9 (Ains), 10 (Acha), and 11 (Parg) suggesting that shorter arrays might be more prone to being lost in the social parasites.

Whether or not this is a consequence of single mutational events deleting entire tandem arrays (particularly in *P. argentina*) or whether successive gene losses reduced OR tandem arrays remains to be studied and is beyond the scope of this study. We have included our analyses of tandem array evolution in the Supplementary Material (newly added section 19) and included a sentence in the manuscript on these findings.

REVIEWER COMMENT:

Line 323, I don't quite follow what data or numbers the authors are referring to when they claim that the changes in inquiline ants are 'on par' to those in intracellular endosymbionts. That is quite the claim without numbers.

RESPONSE:

The changes we refer to here are not with regard to the phenotypic or genomic changes but to the changes in life history. Our reasoning here is that inquiline ants, just like endosymbionts or highly specialized parasites, have completely shifted to a life in extreme proximity to their host.

REVIEWER COMMENT:

Line 125, I think the word 'site' is missing after 'degenerate'

RESPONSE:

Thank you for spotting this. We have corrected this accordingly.

REVIEWER COMMENT:

Line 155, I believe the authors meant to write $dN/dS < 1$ for relaxed purifying selection

RESPONSE:

Exactly! We have changed this accordingly.

REVIEWER COMMENT:

Is the incomplete link in the Supplemental table 29 header supposed to be there? If so, how should the reader access this?

RESPONSE:

No! This has been removed.

REVIEWER COMMENT:

Log spelled differently (lg, log2) in two subplots within Sfig 27

RESPONSE:

We have removed the plot as it was not referred to in the main text. We however updated Sfig 13 (previously Sfig12) where the same mistake occurred. Sfig13 now also reflects the $dN/dS < 10$ cutoff suggested by the Reviewer above.

REVIEWER COMMENT:

Overlapping labels Sfig 28, 30 (offsets or similar could help).

RESPONSE:

We have updated plot 28 and 30 to avoid overlapping labels.

REVIEWER COMMENT:

Figure 2B&C - this might be easier / more interpretable if combined into one, (particularly after addressing issue 3, example B), might also not, but I'd encourage the authors to try and see if it makes sense while simplifying the plot.

RESPONSE:

We would like to point out that Figure 2B shows results from a different analysis (evolutionary rates modelled with absREL) than Figure 2C (evolutionary rates modelled with RELAX).

REVIEWER COMMENT:

Figure 2E - the inset plot is sufficiently similar in zoom to the wider original it doesn't really show us anything not already apparent in the larger plot.

RESPONSE:

We agree that the focus was a bit too wide for the inset plot. We have now adjusted the x-axis range of the inset to -5 to 5.

REVIEWER COMMENT:

Every figure using the viridis color scale. I personally would recommend describing the low color as purple, not blue. It's a bit nit picky but the scale goes through blues, the extreme is the purple.

RESPONSE:

We have changed "blue" to "purple" in the legends of plots using viridis.

REVIEWER COMMENT:

Finally
Thanks, I learned somethings reading the paper, and despite the major issues that need addressing, it is a substantial and interesting work already.
- Dr. Alisandra K Denton

RESPONSE:

Thank you for a thorough and very fair review.

Reviewer #2 (Remarks to the Author):

Review of Schrader et al. (Nature Comm.)

REVIEWER COMMENT:

Overall, this manuscript was a very nice read. The analyses and interpretations are good. My criticisms of the manuscript are about not what is presented, but rather what is not presented (omissions). There are omissions of the relevant literature, there are omissions of analyses that were done, and there are analyses that are not done. These omissions leave the impression that there is a bias in the framing of the manuscript – this is unlikely the intent of the authors, but a byproduct of the publication process (perhaps an over-focus on the story at the expense of the data). At any rate, these omissions are addressable. Not addressing these makes the manuscript both less valuable, but also makes it incomplete and biased.

RESPONSE:

We appreciate that the reviewer considers our study interesting. Unfortunately, we think the criticism of omissions of analyses is unjustified given the space constraints of a *Nature Communications* manuscript. Obviously, and like any other team of researchers, we are focusing on the most relevant findings of our study. However, we do provide extensive Supplementary Materials that also include analyses that could not be included in the main manuscript given the strict word limit. In the main text, we reported the results that offered most novel insight. Regardless, we have now addressed the criticism of omitting analyses by reporting on a number of analyses that did not yield significant results (e.g. lineage-specific genes, gene losses in the host species, etc.). With regard to missing references, we would like to point out that Nature Communications has a soft limit at 70 references, which forced us to remove a number of citations we would have otherwise included.

We have individually addressed each of the reviewer's comments and have modified the manuscript accordingly.

REVIEWER COMMENT:

The largest omissions from the literature are missing citations (either completely lacking or lacking at the appropriate places). There are published studies, some remarkably similar in basic premise to this one, that are not cited. Why? The contrast, for example, between the results of genomic changes in *Pogonomyrmex/Vollenhovia* social parasites relative to their hosts (Smith et al. 2015, *Mol. Biol. Evol.*) with the results of this study are interesting. There are interesting differences in methods and also differences in results – these differences are enriching rather than distracting to the story that is being told.

RESPONSE:

As mentioned above, Nature Communications enforces a limit on the number of references cited, forcing us to remove several references that we would otherwise have included.

We are familiar with the Smith et al. 2015 paper. So far we did not cite this study, because it has the caveat of employing a mapping approach to find differences in the genome sequences between hosts and parasites. By mapping short read data from parasites against the assembled host genome, the authors retrieved the genomic composition of the parasites using the host as reference. This approach is for example quite likely to fail at identifying gene loss in such highly divergent and redundant gene family as the odorant receptors. Further, it prohibits studying structural differences between genomes, which, as we show, can be a relevant factor. Failing to detect a higher frequency of non-sense or frame-shifting mutations in genes in the social parasites, Smith et al. conclude that there is no excessive gene loss in social parasites (without actually testing for gene loss) and that different phenotypes of host and parasite should largely be attributed to gene expression differences (which they did not study). In fact, Supplementary Table S1 of Smith et al. 2015 suggests that there are between 267 to 1800 genes lost in the social parasites relative to their hosts. The experimental design of using mapped short read data to reconstruct the inquiline genomes however makes it impossible to determine whether these are true losses or the consequence of failed mapping.

In contrast, our study reveals marked differences in the coding sequence of genomes of hosts and parasites. Comparing the methodologies of the two studies, we think Smith et al. were unable to detect gene losses and other genome re-arrangements to the same level of detail we did. Therefore, one of the main conclusions in the Smith et al. 2015 article – that social parasite phenotypes are largely explained by regulatory changes, because they did not find increased rates of coding sequence changes and gene loss – is likely based mostly on methodological limitations.

However, we agree with the reviewer that the basic premise of the Smith et al article was similar, and we now cite Smith et al at the beginning of the discussion, so the readers are aware of this related study.

REVIEWER COMMENT:

Omitting mention of these other results is very problematic. There are other, similar papers, that could also be cited much earlier in the manuscript, for example the Cini et al. perspective/theory paper on social parasite

evolution. This paper is referenced in passing much later in the manuscript. Another omission referenced in one of my later comments is a paper by Feldmeyer et al. (2017) in BMC Evol. Biol. – this looks at some genomic changes between slave-making parasites and their hosts.

RESPONSE:

In response to the reviewer's request, we now cite Cini et al in the introduction as well as multiple times in the discussion. The reason why Feldmeyer et al. 2017 was not included in this manuscript is that it is a study on positive selection in slave-making ant species. In contrast, our study examines the role of relaxation of selection in inquiline social parasites. Although, both topics are of course related (social parasites in ants), the differences between our manuscript in methodology, the biology of the study organisms, and particularly the hypotheses that are tested are substantial (Feldmeyer et al used a transcriptomic approach to find candidate genes under positive selection in slave making ants and their hosts). Therefore, given the limited number of references we can include, we prefer to not include this paper.

REVIEWER COMMENT:

The omissions of results are also troubling. The storyline is intensely focused on the changes in the social parasites. This is understandable, but by only reporting on what has changed in the parasites ignores changes in the hosts. For example, families that have high gene losses in social parasites are discussed, but did the hosts also lose genes? By ignoring this symmetry (bifurcation between host/parasite) we lose half the story and bias both the story and the conclusions. For example, what processes (GO terms) are enriched in the genes/families lost by the hosts? Looking at these is essentially a control for looking at what happens in the parasites.

RESPONSE:

It is unfortunate that the reviewer gained the false impression that we did omit results for the sake of a better narrative. Regarding the gene family size analysis, we do report inferred gene loss rates in the hosts in Figure 3a. We have also tested for GO term enrichment of genes lost in the hosts or genes gained in parasites, which did not yield any significant GO terms. Any such results would of course be very interesting and we would gladly have included them in our manuscript. Yet, our analyses produced no such results.

To be more explicit, we have now added a sentence in the result section that no GO terms were enriched among genes lost in the hosts or gained in the parasites, regardless if we did or did not correct for multiple testing (following a suggestion by Reviewer 1). This information is now also included in the Supplementary Material (section 16).

REVIEWER COMMENT:

The omissions of analyses that were not done, or are not reported, are, for example, analysis of lineage-specific genes or analysis of regulatory elements. I do not intend to say that these need to be included, but a large focus on just what is lost (i.e., gene loss) and not what might be gained (novel genes) is a bias. This bias could be addressed in text – contextualize the results. It is ok to say that the analysis was not done, but to not admit that only looking for losses will bias the conclusion toward finding losses is a slight to the reader.

RESPONSE:

The suggestion of willful omission to bias the results is both unfounded and inappropriate. Over the entire course of our study, we have actively searched for adaptive changes associated with social parasitism (see e.g. L175 ff), but we did not detect more than what we report. We also discuss this explicitly in the manuscript (see L372 ff). We also explicitly state that we have not included regulatory, non-coding regions or epigenetic mechanisms and that these deserve further study (L375-377)

We did perform analyses of lineage specific genes and more generally gene gains but did not find any evidence of gains in the parasites that are either convergent or parallel or in any other way clearly relevant to their biology as inquiline parasites. Similar to the analysis of gene gains, lineage specific genes in social parasites did not show any significantly enriched GO terms. We have now added details about lineage-specific genes to the Supplementary Material (page 14ff).

We also performed a number of additional analyses that did not yield any significant results, and we did not report those either. For example, we have spent considerable efforts to explore evolution of regulatory elements using conservation scores from whole genome alignments as implemented in phyloP and Phast. While the general trend was very similar to what we describe in the manuscript (i.e. most dramatic differences in *P. argentina*, little evidence for convergent/parallel changes across all 3 parasites indicative of adaptive evolution), the analyses were simply not sufficiently robust and conclusive to merit inclusion in this manuscript.

To avoid the impression that analyses were not done or results willfully held back, we have now expanded the paragraphs on gene family size evolution analyses in the method section, mentioning GO term enrichment analyses that did not yield any significant results, and added information on this in the Supplementary Material.

We now further write in the result section:

“No GO terms were significantly enriched among genes gained in social parasites or genes lost in the host species.”

REVIEWER COMMENT:

Below are specific comments.

L105-107: It seems a falsehood, at best, to term this manuscript the first to explore molecular evolution of social parasitism. This phrasing does not honor previous studies (some going back decades). Perhaps the authors are relying on the word ‘detailed’ in this sentence, but that is too subjective a word. This is not even the first genomic study on social parasitism. This should just be deleted.

RESPONSE:

We agree that the sentence could leave the false impression that this is the first study molecular evolution of social parasitism. We have hence removed the phrasing “the first”.

REVIEWER COMMENT:

L125: you are missing the word “sites”?

RESPONSE:

Thank you for spotting this. We have corrected this mistake.

REVIEWER COMMENT:

L155: Do you mean $dN/dS < 1$ for purifying?

RESPONSE:

Yes. We corrected the mistake and replaced with $dN/dS < 1$.

REVIEWER COMMENT:

L161: I am not sure the results suggest “erosion” of social parasite genomes. Yes, the results (as presented) suggest that protein coding genes are under less intense selection, relative to non-parasites. So, relative to non-parasites the protein coding portion of the genomes of social parasites is under more relaxed selection --- this is not ‘erosion’. Insufficient information is given to come to the conclusion of ‘erosion’ of the protein coding genome. Erosion makes me think that the genome is ‘falling apart’, but rather it is changing.

RESPONSE:

In the literature, the term “genome erosion” is used in the context of endosymbiont and parasite evolution describing the loss of genes from genomes of endosymbionts or parasites when compared to their free-living relatives. That is exactly analogous to the situation observed in the social parasites of *Acromyrmex* leaf-cutting ants. In addition, the concept of “genetic erosion” is also used in conservation biology/genetics, describing the loss of genetic diversity in the population of an endangered species. This analogy also applies to the ant social parasites because several species have reduced population sizes and suffer from inbreeding and a loss of genetic diversity.

REVIEWER COMMENT:

Additionally, it would be useful to report sample size of genes in each comparison (including as a percent of those annotated or the shared ortholog set used in the analysis) – that is, how many genes in parasite and host lineages have $dN/dS > 1$, etc.? What is the percent of those genes of the total in the dN/dS analysis?

RESPONSE:

The analysis of evolutionary rates summarized in Figures 2B to E are based on initially 7750 single-copy orthologs across all 6 species (6966 after filtering for recombination, as described in Supplementary Material, section 13). We have now implemented another filtering step removing genes with dN/dS estimates > 10 , following a suggestion by Reviewer #1. The final set of genes is hence now 3850 for the analysis underlying Figure 2B, and 3616 for the analysis summarized in Figure 2D. We have added this information to the Figure legend.

We would like to note that the RELAX analysis underlying Figure 2C,D,E is based on analyzing across each gene in the set those regions that evolve on average with $dN/dS < 1$ (Figure 2C), or that evolve on average with $dN/dS > 1$ (Figure 2D). Therefore, 100 % of the genes included in the analysis contribute data to Figures C,D, and E. We have added the information about the number of genes with $dN/dS > 1$ in different branches in the absREL analysis in the Supplementary Material (newly added Supplementary Table 36).

REVIEWER COMMENT:

L167: Report this also as a percent. Presumably it is out of the ortholog set which is 6338, so 4%? This is a lot, I think. Same applies to the next part on intensified selection.

RESPONSE:

The analysis was based on 7750 single-copy orthologs shared among the species included in this analysis (6966 after filtering, see above). 6338 is the number of single-copy orthologs shared across all attines included in the phylogeny. We have now included the percentages of genes showing evidence of relaxed selection (3.34%) or intensified selection (1.46%) in the results.

REVIEWER COMMENT:

L179: I am not convinced that there is evidence of ‘massive genome erosion’ (as previously stated). Furthermore, what genes are under relaxed/intensified selection in (at least one of) the hosts and what are their enriched GO terms? There is a focus on what has changed in the social parasites and nothing about change in the hosts. Without equal attention to both parasite and host it is hard to tell whether the observed changes are unique to the parasites and whether the GO terms make much sense.

RESPONSE:

We have removed the word “massive” in line 179.

The RELAX test compares two sets of branches in a given phylogeny, in our case we compare the socially parasitic branches to the non-parasitic host branches in *Acromyrmex*. Therefore, the changes we report are in the socially parasitic lineages relative to the non-parasitic lineages. With regard to our findings, we do find an excess of genes under relaxed selection (233) over genes under intensified selection (102) in the parasite lineages, thus suggesting that on average selection in the parasites is relaxed compared to the non-parasites (as also demonstrated in the analyses underlying Figures 2B to 2E).

Following the criticism raised here by Reviewer #2 and related criticism by Reviewer #1 requesting corrections for multiple testing in GO enrichment analyses, we have edited the corresponding sections in the results (lines 227ff) to no longer refer to the GO enrichment tests. After correcting for multiple testing, no GO terms are enriched in genes under intensified selection or under relaxed selection in the parasites. We have included this information in the Supplementary Material (page 28).

REVIEWER COMMENT:

L186: An interesting analysis that is almost conspicuously missing is ‘novel’ gene gains. The ortholog set is relatively small (half of the annotations), suggesting that there are a number of lineage specific genes in each of the lineages. These analyses only look at gains/losses in gene families, but this is not likely the whole story. I am not saying that this type of analysis is essential to include, but if it has been done then it should be included. And if it is not yet done then perhaps acknowledging what is unknown is relevant – that is, contextualize your results to say that there are gene losses detected in your analysis, but that your analysis is not comprehensive. By only focusing on what you did find, and ignoring what you didn’t (or didn’t look for) biases your narrative.

RESPONSE:

We have included a very detailed analysis of gene family size evolution to detect changes in gene families across hosts or across parasites. As stated above, we have performed analyses of lineage specific genes in social parasites, but these did not yield any insights worth presenting.

We do not understand why the reviewer implies again that we willfully omitted results that could hint at adaptive changes related to inquiline parasitism. Such results would have been great additions to our manuscript, but we did not obtain them.

We have now added a short paragraph on lineage specific genes in the Supplementary Materials (pages 14ff) stating that no GO terms were significantly enriched among these gene sets and we now also include tables reporting all lineage-specific genes in the three parasites for which homologs in SwissProt could be retrieved.

REVIEWER COMMENT:

L187: 'an' outgroup

RESPONSE:

We corrected the mistake.

REVIEWER COMMENT:

L201: What other families experienced losses? Omissions like this, and the lack of comparative data from hosts as noted above, are useful for story-telling, but do not tell an unbiased story. I continue to get the impression that the authors are more focused on telling a story with a predefined end than putting the data 'on the table'. Here I think that the contraction in ORs is very interesting, but were there other contractions that could also be interesting? Are there contractions that are consistent in the hosts? My guess is probably not, but this is not clearly articulated in the manuscript and so doubts are sown by omissions. ***Note, as I read down in the ms I see the other gene family analyses. I decided to leave in the previous comment because I think it is relevant to address the order of analyses in the manuscript. I suggest that a reference to the other family analyses appears in the first sentence of the OR gene family analysis. Or, a modified version of this whole paragraph goes to the beginning of the section where you begin gene family analysis.

RESPONSE:

The OR family is the only gene family that came up in the GO enrichment analysis of gene family size evolution (see also comments above), and this was reinforced after correcting for multiple testing as suggested by Reviewer #1. Hence, and given we also looked in detail at other gene families as acknowledged by the reviewer, we hope to have persuaded the reviewer that it is not necessary to list all the other gene families that we analyzed already at the beginning of the section on OR genes.

REVIEWER COMMENT:

L351: Similar statements are made by other papers, but are ignored here (see Feldmeyer et al. 2017, BMC Evol. Biol., Smith et al. 2015 Mol. Biol. Evol.). Furthermore, the perspective/theory paper by Cini et al. that is cited is not cited here. These sorts of omissions reflect poorly on this paper.

RESPONSE:

We agree that our phrasing did suggest that the idea of social parasites being loss-of-function models only arose from our analyses. We have now changed the sentence to now state that our study "confirms" that social parasites are such models and now cite Cini et al. and Smith et al. 2015 in this context.

REVIEWER COMMENT:

L441: I am not too sure about precedent for this phrase, but I would recommend a change from 'evolutionary potential' to 'selective potential' to simply reflect that the predominant evolutionary mechanism in question is selection (i.e., a decrease in the potential for selection, relative to drift, when N_e is low).

RESPONSE:

Previous authors made the argument that inquiline social parasites are inbred and therefore should be prone to extinction. In those studies, which are referenced in the text, the authors suggest that social parasites have a reduced "evolutionary potential". In the context of this discussion, we believe using the term evolutionary potential remains correct.

REVIEWER COMMENT:

L447: This language could be in the Smith et al. (2015) paper that looked at genomic change between social parasites and hosts, too. Again, the omission of comparing the results of these papers is a major omission.

RESPONSE:

We are not certain which argument the reviewer is making here, but we hope by citing Smith et al twice in our manuscript now, we could resolve the impression of omission.

Reviewer #3 (Remarks to the Author):

In this paper the authors investigate whether the evolution of parasitism is associated with a decrease of the strength of selection on some gene families. The study reveals some interesting pattern despite the fact that there are only 3 parasitic species (and 2 events of evolution of parasitism). The authors characterize global patterns of relaxed constraint, and then move on to focus on the large olfactory receptor family, noting large reductions. microCT were also used to ask if olfactory lobe has changed size in a correlated way. The authors end the paper on a quantification of different mutational classes (inversions, transpositions, deletions), showing largely consistent results – increased mutational events along parasite branches (with some exceptions).

Overall, the results of the paper are quite interesting and of general interest, and the data seem of high quality. We do, however, have concerns/questions regarding a number of the analyses.

REVIEWER COMMENT:

1) An important issue is whether the patterns uncovered are due to changes in social behaviour per se or whether it is due to a decrease of N_e (which is expected to occur with the evolution of social life). Surprisingly, the authors assess changes of N_e for only 2 species. It would be useful to do this for all the studied species (and also compare the data with other published work -see Romiguier et al. JEB 2014.).

RESPONSE:

We agree with the reviewers that it would be useful to expand our analysis of N_e to all studied species. The analysis we performed requires resequencing data of at least two unrelated individuals from the same population. Unfortunately, we were so far not able to collect appropriate samples of the two remaining social parasite species *A. charruanus* and *P. argentina*. We have now added a sentence in the main text to highlight this limitation. We have further included a reference to Romiguier et al. in the concluding paragraphs of our introduction, to emphasize that N_e is expected to be low in social insects to begin with. We would however like to point out that Romiguier et al. did not directly estimate N_e in social and solitary species, but compared the effective selection coefficient ($S=4N_e s$) and concluded that S being lower in social species than non-social is consistent with a lower N_e in the former. Hence, it is difficult to directly compare their results to ours.

REVIEWER COMMENT:

2) The authors discuss various "selection" analyses throughout the paper but need to state more precisely the type of selection the test are investigating. For example, it is not clear what is meant w/ "relaxed positive selection" (Fig2D) using dN/dS values. Similarly, on L175, "intensified selection" could mean several things (but positive selection is what the authors are testing for), and on L218 "relaxed selection" should be "relaxed purifying selection".

RESPONSE:

We have expanded the figure legend of Figure 2 to include more precise statements about the different types of selection addressed in panels C and D. Relaxed positive selection refers to cases where dN/dS is > 1 across all species, but closer to 1 in the parasite compared to the background branches. Intensified selection is in fact the correct term in L175 as it refers to both, intensification of positive selection (i.e. $dN/dS > 1$ and dN/dS test-set $> dN/dS$ reference-set) and intensification of purifying selection (i.e. $dN/dS < 1$ and dN/dS test-set $> dN/dS$ reference-set). This terminology was established in the original publication of the RELAX model (Wertheim et al. 2015). We have expanded the method section outlining the RELAX model in greater detail.

Regarding the expression in L218 the reviewers are correct. We have changed this accordingly to "relaxed purifying selection".

Wertheim, J. O., et al. (2015). "RELAX: detecting relaxed selection in a phylogenetic framework." *Mol Biol Evol* 32(3): 820-832.

REVIEWER COMMENT:

3) Could the authors also investigate codon usage bias as a way of detecting relaxed purifying selection? In addition, linkage disequilibrium is expected to be increased in bottlenecked populations, so perhaps this would be an additional line of evidence, and another interesting analysis to contrast with estimates of N_e .

RESPONSE:

Following the reviewer's suggestion, we have explored codon usage bias (CUB) of hosts and parasites for CDS sequences of single-copy-orthologs across species. We compared codon usage bias in detail in each of the five *Acromyrmex* and *Pseudoatta* species (*A. echinator*, *A. insinator*, *A. heyeri*, *A. charruanus*, *P. argentina*) using ENC-GC3 plots and neutrality plots. In general, the

extent of codon bias was high and also highly correlated across species, with differences largely explained by their phylogenetic relationship. The neutrality plots and ENC~GC3 plots suggest that codon usage bias is dominated by natural selection and less by mutational bias in these species regardless of being a social parasite or not. However, we do find that the relationship of ENC to GC3 is (not significantly) closer to the neutral expectation (see Wright et al 1990) in the three parasite species relative to their respective host. Further, correlation coefficients of the neutrality plots are marginally closer to 1 in all parasites relative to their respective hosts. Under strict neutrality, the correlation coefficient is 1, suggesting that GC1 and GC2 evolve relatively more neutral in the social parasites than the hosts.

A comparison in different *Pseudomyrmex* species found a very similar pattern, with CUB being stronger, but not significantly stronger, in mutualistic than in non-mutualistic species (Rubin & Moreau, 2016). This likely indicates that CUB is generally highly conserved among ants.

We have now added a sentence to the result section stating that codon usage is less biased in parasites, but that the signal is much weaker than in the analysis of synonymous and non-synonymous sites. We also refer to the newly added section (Section 14) in the Supplementary Material that summarizes our analysis of CUB and have included a brief summary of our approach to study CUB in the Method section of the main text.

Regarding linkage disequilibrium, we agree with the reviewer that it would be interesting to test whether parasite populations have increased LD. However, with the current single individual resequencing data set of 2 samples each from the Aech/Ains host-parasite pair, we cannot robustly infer LD. In addition, it is not feasible to collect several unrelated individuals of the two other parasite species given their extreme rarity.

Conducting more population genomic analyses are beyond the scope of this paper, given the focus on comparative genomics across species. However, we are already working on a follow up study to analyze a large population genomic dataset for *A. echinator* and *A. insinator*. Based on preliminary results of this follow up study, we can confirm that genetic diversity is extremely reduced in the social parasite, in accordance with a dramatically reduced effective population size.

Rubin, B. E. and C. S. Moreau (2016). Comparative genomics reveals convergent rates of evolution in ant-plant mutualisms. Nat Comm 7: 12679.

REVIEWER COMMENT:

4) For inference of N_e using 2 genomes, it would be helpful to have a citation of the method in the sentence (L145). See also study of Romiguier et al. mentioned above.

RESPONSE:

We have included the following reference to the Method section and the Supplementary:
Schiffels S, Wang K. (2020). MSMC and MSMC2: The Multiple Sequentially Markovian Coalescent. Methods Mol Biol. 2090:147-166. http://doi.org/10.1007/978-1-0716-0199-0_7

REVIEWER COMMENT:

5) The authors do not discuss the results of the MSMC2 analyses much beyond the smaller N_e in *A. ins* compared to *A. ech*. What do the authors make of the inferred recent increase in N_e , that puts the value above the initial (though uncertain) starting N_e ? This should be extended with data from the other species.

RESPONSE:

As outlined above, we could unfortunately not perform the analysis for other host/parasite pairs because we did not have the appropriate duplicate samples. We have now expanded the figure legend to offer a tentative explanation for the recent increase in N_e for both Panamanian species.

*“The recent increase in N_e estimated for both species could be a consequence of deforestation in Gamboa, Panama over the last centuries, as *A. echinator* prefers open habitats over dense forests. Increased habitat availability for the host could in turn also favor growth of the social parasite population.”*

REVIEWER COMMENT:

6) Fig 2C and 2D: These two panels are not clear. Is panel E just a continuation of panel C, and if so, why are there few data points for values just smaller than 1 and many data points for values just greater than 1.

RESPONSE:

We agree we should have made this clearer. Figure 2C summarizes those regions within the single copy orthologs that are inferred to evolve under purifying selection (i.e. $dN/dS < 1$, i.e. regions under purifying selection, so usually $dN/dS \ll 1$, which explains why there are so few values just under 1). Figure 2D focuses on different

regions, i.e. those that evolve at a $dN/dS \geq 1$. Hence, panel D is not a continuation of C but summarizes selectively those regions with $dN/dS \geq 1$. See also Wertheim et al. 2015. To explain the approach better, we have (as also mentioned above) now expanded the method section on the RELAX analysis and also the Figure legend.

Wertheim, J. O., et al. (2015). "RELAX: detecting relaxed selection in a phylogenetic framework." *Mol Biol Evol* 32(3): 820-832.

REVIEWER COMMENT:

7) Fig2E: Not sure much is gained by the inset, but adding a carpet where data point lie would be helpful. It would also be helpful for readers unfamiliar with the RELAX parameters being estimated to have a simpler way of displaying the meaning of x-axis of Fig2E.

RESPONSE:

We have added a carpet and narrowed the focus of the inset (following a suggestion by Reviewer #1). Also, we have changed the x-axis label to "selection intensity k (\log_2)" to give a clearer meaning to k .

REVIEWER COMMENT:

8) L159: It would be helpful to have numerical summaries of genes that fall into the "minority" and "majority" categories.

RESPONSE:

Of the 3616 genes we tested after removing genes with dN/dS estimates > 10 , as suggested by Reviewer #1, we found:

792 genes with $\log_2(k)$ smaller -2
2186 genes with $\log_2(k)$ between -2 to 0
563 genes with $\log_2(k)$ between 0 to 2
75 genes with $\log_2(k)$ larger 2

We have now included this information in the Figure legend.

REVIEWER COMMENT:

9) The interpretation from the GO analyses are not particularly convincing (here and in the discussion). These categories are broad enough to construct many stories.

RESPONSE:

We agree with the reviewers. Also following suggestions by Reviewer #1 to implement correction for multiple testing, we have edited the sections accordingly and no longer refer to the results of the uncorrected enrichment tests.

We now write:

"We identified 233 genes (3.34% of the tested genes) showing evidence of relaxed selection in at least one of the social parasite branches (at $FDR < 0.1$ and relaxation parameter $k < 1$) among single-copy orthologs (see Supplementary Figures 14-16, Supplementary Table 34). However, we also detected signatures of intensified selection (at $FDR < 0.1$, $k > 1$) in 102 genes (1.46%, Supplementary Figures 17-19, Supplementary Table 35), suggesting that the convergent niche shifts to social parasitism also triggered adaptive changes within an overall context of more genomic erosion."

REVIEWER COMMENT:

10) The authors did not find accelerated gene loss in *A. char*, but do not really discuss this much. Why would gene loss be stopped along only this lineage? This is where an N_e estimate for this genome might be helpful. (see comment 15 also)

RESPONSE:

We addressed the differences between *A. charruanus* and *P. argentina* in L410ff, focusing primarily on the rapid changes in *P. argentina*. We have now added two sentences in the discussion to offer explanations for the slower rate in *A. charruanus*. Our working hypothesis is that *A. charruanus* has quickly evolved to a fitness landscape plateau, allowing it to persist without much further change. It is also possible that population level mechanisms such as introgression or hybridization with the host are more prevalent in *A. charruanus* than in *P. argentina*. Regardless of the molecular mechanisms, our findings are in line with the fact that morphologically *A. charruanus* is much less derived from its host than *P. argentina*. Why this is the case remains to be explored. Given we

cannot estimate N_e nor do population-level studies of the species in the field (for reasons outlined above), we have emphasized that the explanations that we offer are speculative.

We now write in the discussion:

*“It is possible that *A. charruanus* quickly reached a stable fitness plateau after gene flow with ancestral *Pseudoatta* had ceased, or that new hybridization/introgression with its host somehow constrained further genome evolution, but any such considerations remain speculative with the currently available data”*

REVIEWER COMMENT:

11) The results on OR losses are interesting. More details/quantifications about number of full/large deletions, nonsense mutations etc. would be interesting. It has been shown in *Drosophila* that there is some type of ‘readthrough’ of OR nonsense mutations (and maybe other genes), and thus that pseudogenes can’t be annotated alone by this (<https://www.nature.com/articles/nature19824>). The mechanism remains unclear, so it may/may not persist across more distant taxa.

RESPONSE:

“pseudo-pseudo” ORs should have premature stop-codons but conserved coding sequence following the premature stop. In the process of annotating ORs in the attine genomes, we did not encounter any such case. We thus have no evidence to suspect that stop readthrough occurs in ORs in this lineage.

As summarized in Table 2 (see also details in the Supplementary Material we provide), we find more complete ORs in the non-parasitic species, but in fact also find more incompletely annotated (or putatively pseudogenized) ORs in the non-parasitic hosts, rendering the analysis of pseudogenes in the OR family inconclusive.

We now have further compared the genomic organization of ORs (following also a suggestion by Reviewer #1) and found that fewer tandem arrays (with 3 or more genes) are retained in the parasitic species, but that the remaining arrays are on average slightly larger in the parasitic species.

Based on our whole genome alignments, we compared orthologous arrays with regard to their gene count in hosts and parasites. Generally, gene count is conserved in only a few arrays across all species, regardless of the host-parasite angle. It is however striking that *Parg* has the lowest gene count in most of the arrays, as expected by the largely reduced OR gene set in this species.

Together we thus suspect that full gene deletions and faster decay of gene arrays are the dominant forces underlying OR loss in parasites, potentially as a consequence of non-homologous recombination (as already discussed in L399 ff). Another mechanism creating gene deletions could be aberrant transposition of TEs.

Intriguingly, upon manual inspection of OR arrays, we found several cases, (prominently in *Pseudoatta*) where a Mariner transposon had invaded, creating a conspicuous gap in the array.

We have added those additional analyses of OR arrays to the Supplementary Material (Section 19) and refer to them in the main text, where we write in the results:

“In accordance, fewer large OR gene arrays (> 8 genes) were retained in parasites compared to hosts (Supplementary Material).”

REVIEWER COMMENT:

12) L208 and L251: Not sure that that ORs losses should be called “convergent” unless these were independent losses of orthologs. It would be more appropriate to use a term such as “parallel losses of OR family members”, or something along these lines.

RESPONSE:

We agree with the reviewer and have changed “convergent” to “parallel” in the sentence.

REVIEWER COMMENT:

13) One concern with the OR work is that the authors study this gene family (and a few more more such as GRs, MRJP, CPR) in more careful detail than other families. It would be useful to have a balanced gene family-wide analysis. Though it seems like ORs are real outliers and present very interesting evolutionary dynamics, the special treatment of this family does leave open the possibility for some selection bias.

RESPONSE:

We agree with the Reviewer that the focus on ORs would be problematic if this gene family had been picked without any primary evidence that justified such a focus. However, we studied ORs in detail because it was the only gene family that emerged from our GO enrichment analysis of gene families undergoing parallel changes in the social parasites (the CAFÉ analysis, L185ff). In L201ff we wrote that enrichment analyses revealed that “gene loss had most significantly affected genes with olfactory receptor (OR) functions.” After implementing correction for multiple testing as suggested by Reviewer #1, this outcome is even more striking.

REVIEWER COMMENT:

14) The way the hypothesis on L253 is started off is a bit misleading: number of glomeruli does not necessarily have to relate to overall OL size. The count of specific glomeruli is a different than the volumetric measure obtained with the microCT scans (at least as I understand it). Also it would probably be more honest to call the result from these scans as marginally “non-significant” (L260).

RESPONSE:

We agree with the reviewer that glomeruli count does not necessarily correlate with OL size. We have changed the sentence starting in L 263 to “*Previous studies have suggested that during ant evolution the expansions in size and glomeruli number of the olfactory lobes have remained correlated with the expansion of the OR gene repertoire.*” and now cite Tribble et al. 2017 in addition to McKenzie et al. 2016. In the 2017 paper, Tribble et al. show that disrupting OR function by *orco* knockout dramatically reduces the OL size in the clonal raider ant. Together with the study by McKenzie, these findings suggest that the relatively large size of OLs in ants is a consequence of the expanded OR repertoire, which has led us to hypothesize that OLs are reduced in social parasites.

We further agree that it is more honest to call the statistical results marginally non-significant and have changed this accordingly.

Tribble, W., et al. (2017). “*orco* Mutagenesis Causes Loss of Antennal Lobe Glomeruli and Impaired Social Behavior in Ants.” *Cell* 170(4): 727-735 e710.

REVIEWER COMMENT:

15) For the genome rearrangements section, do the authors have any way to gauge the relative confidence in the annotation across mutational classes? Are deletions harder to identify, for example than inversions? *D. cha* stands out again as having a paucity of deletions. In that respect, having details about the mutational events removing OR could be helpful. Also, it would seem possible that the lack of deletions in *D. cha* is related to the relative lack of gene loss (as shown in Fig3A). If not mistaken, this seems worth discussing.

RESPONSE:

We expect that deletions are the easiest to detect, followed by inversions and then translocations, but we have no means of objectively quantifying the confidence across mutational events and classes from our whole genome alignments. As a consequence, we therefore only compared relative rates for each mutational class in the manuscript. In absolute numbers, we find more deletions than inversions or translocations across all branches and species.

Our analyses of OR gene arrays (see above) highlighted the relevance of deletions in the evolution of ORs, but it is difficult to assess the impact of inversions and translocations on this rapidly changing gene family. It is possible that nonhomologous recombination events have induced inversions and/or translocations (as well as duplications for that matter), but with our current data, we cannot explore the impact of each mutational class in greater depth.

REVIEWER COMMENT:

16) Regarding the rearrangements, how many genes are involved/associated in these events?

RESPONSE:

It is difficult to robustly assess the impact of individual mutations (regardless of the mutational class we look at), because of the incontinuity of whole genome alignments in ants, as – from what we know so far – ant genomes have been reshuffled to an extraordinary degree during evolution. Thus, inferring orthologous and syntenic regions in genomes is much more challenging than in e.g. mammals (where genomes are also much larger), particularly with second generation sequencing assemblies as we present here.

Given these limitations, we kept the analysis of our genome alignment in the main text at the current relative scale and did not include an analysis of the genes affected by mutational events. We have however performed the gene level analysis for the sake of the review and now also report that analysis in the Supplementary Material (Section 25), while at the same time emphasizing the uncertainty of these findings.

To quantify the number of genes affected by each mutational class, we first inferred coordinates of transpositions, inversions and deletions for the three parasites, the two hosts and the reconstructed genome of the *Acha/Parg* stem group (N5). We then identified overlap between genes and rearrangements in each genome (see Supplementary Material section 25 for details).

Note that estimating overlap of deletions required the inference of gene coordinates in the ancestral reconstructed genome (i.e. leftover of gene annotations from e.g. *Acha* to the reconstructed genome at N5), rendering these

estimates more unreliable. Similarly, estimates for inversions, and transpositions at N5 are based on gene coordinates inferred from *A. charruanus*. For deletions at N5, we inferred gene coordinates at the Acha/Parg/Ahey stem group, using a liftover from Ahey gene annotations. As shown below, using gene coordinate liftover from a related species had strong effects on the inferred number of affected genes.

Relative to a species' own gene coordinates, we found between 0 (in Ahey) to 7 (in Ains) genes affected by inversions, between 6 (Ahey) to 169 genes (N5) affected by transpositions, and 0 (Ahey) to 22 genes (N5) affected by deletions. When using gene coordinates created by liftover from a sister species, we found substantially more genes affected by deletions (between 553 in Acha to 1509 in Parg). The numbers were also very different for transpositions (6 to 131 genes affected) and inversions (0 to 12 genes affected). Please refer to the Supplementary Material, Section 25 for details.

REVIEWER COMMENT:

17) L311: not sure that it was really gene-family specific. There were multiple gene families in the GO list (table S40), and ORs were on the top but this does not mean it was specific – just enriched.

RESPONSE:

We have changed “gene-family-specific” to “gene-family-wide”. Note that after correcting for multiple testing, only GO terms related to olfaction were significant (see newly added Supplementary Table 45).

REVIEWER COMMENT:

18) An idea: might it make more sense to keep the “Genome rearrangements” section together with the section “genomic evidence of relaxed selection”? It seems that these data got together, and it reads a bit odd to return to mutation types after going through the olfactory sections.

RESPONSE:

Our motivation to analyze genome rearrangements and synteny decay was that we found increased rates of gene loss in the social parasites (see line 283). We thus prefer to keep the section on genome rearrangements after the description to gene family changes, to preserve this line of thought.

REVIEWER COMMENT:

18) L413: deletions do not look to be increased in common ancestor, right?

RESPONSE:

We have now specified this to read: “... *the frequencies of genome rearrangements (transpositions) were increased in the common ancestor of ...*”.

REVIEWER COMMENT:

19) L125: typo in: “..4-fold degenerate inferred...”

RESPONSE:

We have corrected this.

REVIEWER COMMENT:

Roman Arguello and Laurent Keller

RESPONSE:

Thank you for this constructive review

REVIEWER COMMENTS

Reviewer #1 (Remarks to the Author):

Thank you to the authors for investing the time and effort to respond thoroughly and satisfactorily to my concerns. The manuscript is clearer, more complete and more convincing for the modifications.

One minor thing that I caught during the re-read.

On e.g. line 200 (but also throughout wherever p or FDR values are mentioned).

Should this not read "FDR=5.6e-6"? Generally one would use '<' for an alpha threshold (e.g. for a statement like "N genes showed significant differences with FDR < 0.05"), but precisely report the p or FDR value with an "=" when talking about a single specific result.

- Dr. Alisandra K Denton

Reviewer #2 (Remarks to the Author):

The revisions done by Schrader et al. are comprehensive. The authors have done a good job addressing the critiques of all reviewers.

Reviewer #3 and 4 (Remarks to the Author):

In the same order as our previous comments:

1. It is unfortunate that the authors could not apply the same method for inferring N_e to more than the two lineages presented in Figure 2, which relied on two genomes. What about applying the method of Li and Durbin's to their single genome (Nature 2011, Inference of human population history from individual whole-genome sequences)? This would allow the authors to include N_e estimates for each of the genomes that they have, as well as other published genomes (as suggested in our initial review). Given the importance of N_e to the author's story, this still seems to be an important point to address.

2.Sufficiently addressed.

3.Sufficiently addressed.

4.Sufficiently addressed.

5.see comment to point #1, above.

6. The additional details and changes have clarified the figure. However, we were not convinced by the speculation that: "The recent increase in N_e estimated for both species could be a consequence of deforestation in Gamboa, Panama over the last centuries, as *A. echinatio* prefers open habitats over dense forests. Increased habitat availability for the host could in turn also favor growth of the social parasite population."

This seems like too short of a timespan for such a change to occur.

7. Sufficiently addressed.

8. Sufficiently addressed.

9. The section is improved although their evidence is still very weak.

10. The new speculations are not really fully convincing. Will be interesting to see what the N_e estimates are.

11. Sufficiently addressed.

12. If the authors agreed w/ the change from convergent to parallel for this sentence, then the switch should be applied consistently throughout the manuscript.

13. Sufficiently addressed.

14. Sufficiently addressed.

15. These analyses could be limited by relative differences in genome quality (and the overall fragmented nature of their assemblies). This is a weakness, but the results are also in the same direction as the other nucleotide-based analyses, which is helpful.

16-19. Sufficiently addressed.

Laurent Keller, Roman Arguello

Reviewer #1 (Remarks to the Author):

REVIEWER COMMENTS:

Thank you to the authors for investing the time and effort to respond thoroughly and satisfactorily to my concerns. The manuscript is clearer, more complete and more convincing for the modifications.

One minor thing that I caught during the re-read.

On e.g. line 200 (but also throughout wherever p or FDR values are mentioned). Should this not read "FDR=5.6e-6"? Generally one would use '<' for an alpha threshold (e.g. for a statement like "N genes showed significant differences with FDR < 0.05"), but precisely report the p or FDR value with an "=" when talking about a single specific result.

- Dr. Alisandra K Denton

RESPONSE:

We have changed "<" to "=" in lines 201, 231, 242, 265, 299, 800 and 801. Thank you once again for the thorough review.

Reviewer #2 (Remarks to the Author):

REVIEWER COMMENTS:

The revisions done by Schrader et al. are comprehensive. The authors have done a good job addressing the critiques of all reviewers.

RESPONSE:

Thank you for your critical review of our work.

Reviewer #3 and 4 (Remarks to the Author):

REVIEWER COMMENTS:

In the same order as our previous comments:

1. It is unfortunate that the authors could not apply the same method for inferring N_e to more than the two lineages presented in Figure 2, which relied on two genomes. What about applying the method of Li and Durbin's to their single genome (Nature 2011, Inference of human population history from individual whole-genome sequences)? This would allow the authors to include N_e estimates for each of the genomes that they have, as well as other published genomes (as suggested in our initial review). Given the importance of N_e to the author's story, this still seems to be an important point to address.

RESPONSE:

The suggested method by Li & Durbin (2011) requires unphased sequencing data from a single diploid individual and cannot be applied to assemble genomes, similar to the MSMC2 method that we apply. Unfortunately, such data is not available for any of the other social parasites or host lineages included in our study nor for any other leaf-cutting ant species for which genomes have been published before.

In general, including even more distantly related species will introduce a much higher degree of uncertainty in the modelling, which we can so far exclude in our focal comparison of two sympatric, recently diverged species.

However, to strengthen our argument, we have expanded our demographic modelling to include *Acromyrmex octospinosus*, which is the closest relative to the *A. insinuator/A. echinator* clade and occurs in sympatry in Gamboa, Panama (Schultz et al. 1998). Our 3-species model suggests that the demographic history of *A. octospinosus* is very similar to that of *A. echinator*, except for an earlier onset of population expansion in the former. Most importantly, the effective population size of *A. insinuator* is consistently smaller at any time point compared to both non-parasitic species. We have updated the corresponding sections in the manuscript accordingly.

We now write in the results section:

Coalescent analyses on one host-parasite pair (A. echinator and A. insinator) and a third, non-parasitic, closely related species (A. octospinosus) for which the appropriate samples (two individuals from different colonies) could be collected, confirm that Ne was indeed consistently smaller in the social parasite compared to its free-living relatives (Figure 2A).

The legend for Figure 2A now reads:

(A) Effective population size (Ne) of the social parasite A. insinator is consistently smaller than of its close relatives A. echinator and A. octospinosus.

We have further updated the corresponding sections in the Method section and the Supplementary Material.

Schultz, T. R., et al. (1998). "Acromyrmex insinator new species: an incipient social parasite of fungus-growing ants." INSECTES SOCIAUX 45(4): 457-471.

REVIEWER COMMENTS:

2. Sufficiently addressed.
3. Sufficiently addressed.
4. Sufficiently addressed.
5. see comment to point #1, above.

RESPONSE:

In response to comment 6. below and given we cannot expand the analysis of demographic history to the other parasite lineages, we propose to avoid speculating about the nature of the recent increase in Ne.

REVIEWER COMMENTS:

6. The additional details and changes have clarified the figure. However, we were not convinced by the speculation that: "The recent increase in Ne estimated for both species could be a consequence of deforestation in Gamboa, Panama over the last centuries, as A. echinator prefers open habitats over dense forests. Increased habitat availability for the host could in turn also favor growth of the social parasite population."

This seems like too short of a timespan for such a change to occur.

RESPONSE:

We have removed the sentence from the legend.

REVIEWER COMMENTS:

7. Sufficiently addressed.
8. Sufficiently addressed.
9. The section is improved although their evidence is still very weak.
10. The new speculations are not really fully convincing. Will be interesting to see what the Ne estimates are.

RESPONSE:

We cannot at this point provide further evidence against or for our speculation, which we label as such in lines 422: "but any such considerations remain speculative with the currently available data."

REVIEWER COMMENTS:

11. Sufficiently addressed.
12. If the authors agreed w/ the change from convergent to parallel for this sentence, then the switch should be applied consistently throughout the manuscript.

RESPONSE:

We agree that the losses of specific ORs initially referred to in this comment should not be labelled “convergent”, because we do not show these losses to be independent losses of orthologous ORs. However, we are convinced that using the term convergent (e.g. “convergently evolved signatures of genome-wide and trait-specific genetic erosion”) in other contexts in our manuscript is justified, because we do have two independent origins of social parasitism with both showing similar genomic and phenotypic changes.

REVIEWER COMMENTS:

13. Sufficiently addressed.

14. Sufficiently addressed.

15. These analyses could be limited by relative differences in genome quality (and the overall fragmented nature of their assemblies). This is a weakness, but the results are also in the same direction as the other nucleotide-based analyses, which is helpful.

RESPONSE:

We agree with the reviewers that the analysis of OR gene arrays is only useful as supporting evidence and would not be convincing on its own.

REVIEWER COMMENTS:

16-19. Sufficiently addressed.

Laurent Keller, Roman Arguello

RESPONSE:

Thank you for the insightful review of our work.

REVIEWERS' COMMENTS

Reviewer #4 (Remarks to the Author):

I appreciate the authors efforts to address our questions/comments. All points have now been addressed.

Reviewer # 4 (Remarks to the Author):

REVIEWER COMMENTS:

I appreciate the authors efforts to address our questions/comments. All points have now been addressed.

RESPONSE:

Thank you for your critical evaluation of our work.